METHODS AND RESOURCES

# Identification of a highly efficient chloroplast-targeting peptide for plastid engineering

**Chonprakun Thagun[1,2], Masaki Odahara[3], Yutaka Kodama[2,3]\*, Keiji Numata [1,3]\***

**1** Department of Material Chemistry, Graduate School of Engineering, Kyoto University, Kyoto-Daigaku-Katsura, Kyoto, Japan, **2** Center for Bioscience Research and Education, Utsunomiya University, Tochigi, Japan, **3** Biomacromolecules Research Team, RIKEN Center for Sustainable Resource Science, Saitama, Japan

\* kodama@cc.utsunomiya-u.ac.jp (YK); keiji.numata@riken.jp (KN)

## Abstract

Plastids are pivotal target organelles for comprehensively enhancing photosynthetic and metabolic traits in plants via plastid engineering. Plastidial proteins predominantly originate in the nucleus and must traverse membrane-bound multiprotein translocons to access these organelles. This import process is meticulously regulated by chloroplast-targeting peptides (cTPs). Whereas many cTPs have been employed to guide recombinantly expressed functional proteins to chloroplasts, there is a critical need for more efficient cTPs. Here, we performed a comprehensive exploration and comparative assessment of an advanced suite of cTPs exhibiting superior targeting capabilities. We employed a multifaceted approach encompassing computational prediction, in planta expression, fluorescence tracking, and in vitro chloroplast import studies to identify and analyze 88 cTPs associated with *Arabidopsis thaliana* mutants with phenotypes linked to chloroplast function. These polypeptides exhibited distinct abilities to transport green fluorescent protein (GFP) to various compartments within leaf cells, particularly chloroplasts. A highly efficient cTP derived from *Arabidopsis* plastid ribosomal protein L35 (At2g24090) displayed remarkable effectiveness in chloroplast localization. This cTP facilitated the activities of chloroplast-targeted RNA-processing proteins and metabolic enzymes within plastids. This cTP could serve as an ideal transit peptide for precisely targeting biomolecules to plastids, leading to advancements in plastid engineering.

## Introduction

Plastids are DNA-containing endosymbiotic organelles that are the sites of photosynthesis and supply chemical energy to plant cells. Plastids contain plastidial proteins encoded by numerous circular plastid-genome DNAs (plastomes) and by DNA from the nuclear genome [1]. Approximately 120 of these plastidial proteins are encoded by the plastome, whereas the majority are encoded by the nuclear genome [2]. Nucleus-encoded plastidial proteins are synthesized as preproteins (i.e., precursors) in the cytosol and posttranslationally imported into plastids and subplastidial compartments such as the thylakoids [3]. In

**Funding:** K. N. was financially supported by the Ministry of Education, Culture, Sports, Science and Technology (MEXT), Data Creation and Utilization-type MaTerial R&D project: https://dxmt.mext.go.jp/en/about; Grant Number JPMXP1122714694, Japan Science and Technology Corporation, Exploratory Research for Advanced Technology (JST-ERATO): https://www.jst.go.jp/erato/en/; Grant Number JPMJER1602, and Japan Science and Technology Corporation, Program on open innovation platform for industry-academia co-creation (COI-NEXT): https://www.jst.go.jp/pf/platform/; Grant Number JPMJPF2114. The funders had no role in study design, data collection and analysis, decision to publish, or preparation of the manuscript.

**Competing interests:** The authors have declared that no competing interests exist.

**Abbreviations:** BSA, bovine serum albumin; CBB, Coomassie brilliant blue; cDNA, complementary DNA; CLSM, confocal laser-scanning microscopy; cTP, chloroplast-targeting peptide; EV, empty vector; GFP, green fluorescent protein; HAI, hours after infiltration; HRP, horseradish peroxidase; MEP, methylerythritol 4-phosphate; mTP, mitochondrion-targeting peptide; OD, optical density; qRT-PCR, quantitative reverse-transcription PCR; ROI, regions of interest; TALEN, transcription activator-like effector nuclease; tTP, thylakoid-targeting peptide; VIGS, virus-induced gene silencing; ZFN, zinc-finger nuclease.

general, preprotein import to plastids is directed by transit peptides known as chloroplast-targeting peptides (cTPs) located at either the amino (N)- or carboxyl (C)-terminus of the protein [4,5]. After successful translocation into plastids, the cTP domain is irrevocably excised by plastidial signal peptidases, leaving a mature protein to function in its related organellar process [4–6].

cTPs regulate both the efficiency and specificity of preprotein import into plastids. N-terminal cTPs typically encompass the first 20 to 100 amino acids of the preproteins [7]. These cTPs do not exhibit distinctive characteristics except for the presence of an abundance of hydroxylated serine/proline residues [4,5]. Cytosolic scaffold proteins such as 14-3-3, HSP70, and HSP90 stabilize unfolded preproteins by interacting with amino acids in the cTP domains that are in the appropriate conformation [8]. Once preproteins reach the plastid envelope, cTPs guide preproteins into plastids via an energy-dependent process. This import is mediated by translocons located on the outer and inner envelope membranes (TOC/TIC complexes). Subsequently, stromal processing peptidases remove the cTP portions of preproteins that function within the plastid stroma (the protein-rich fluid inside plastids) at specific cleavage sites [6]. Additionally, various plastidial proteins are transported to the thylakoid membrane and lumen. These thylakoid-localized proteins normally contain bipartite chloroplast/thylakoid-targeting sequences in tandem with 2 distinct cleavage sites [9,10]. The mutation and depletion of cTP domains lead to the mislocalization and malfunctioning of plastidial proteins in plant cells [11–13].

In the field of plastid biotechnology, various cTPs have been employed to direct the localization of recombinantly expressed cargo proteins, such as green fluorescent protein (GFP) [14,15], metabolic enzymes [16–18], protective agents [19], plastid-membrane transporters [20], and RNA-processing proteins to plastids [21]. Moreover, the integration of cTPs into plastid-targeted DNA-editing proteins such as zinc-finger nuclease (ZFN), transcription activator-like effector nuclease (TALEN), and CRISPR-type deaminase has led to selective plastome modifications in plants [22–25]. cTPs derived from plant Rubisco small subunit proteins (RbcS) have been used as biorecognition modules on nanoparticles to specify biomolecule delivery to chloroplasts [26]. In addition, manipulating peptide sequences improved the targeting abilities of cTPs [13,18]. Until now, the comparison of import efficiencies among numerous cTPs has been experimentally performed using various techniques, including in vitro import assays, in vivo protoplast transfection, and expression studies in transgenic plant cells [27,28]. Thus, superior cTPs from available genetic resources remain to be identified and analyzed.

In this study, we classified the plastid-targeting efficiencies of transit peptides derived from nucleus-encoded proteins related to chloroplast phenotypes in *Arabidopsis* (*Arabidopsis thaliana*) mutants [29,30]. We fused computationally predicted cTPs to GFP and compared the plastid-targeting abilities of 89 different cTP-GFPs by in planta expression analysis. Surprisingly, the cTP-GFPs localized to distinct cellular compartments, including chloroplasts, mitochondria, and the cytosol, regardless of their predicted subcellular localizations. Our data emphasize the importance of cTP length and cleavage sites in determining the efficiency of protein translocation into chloroplasts. Among the cTPs, a superior cTP that showed approximately 10 times higher chloroplast-targeting efficiency than the widely used *Arabidopsis* AtRbcS cTP was identified. This outstanding cTP exhibited a greater ability than native cTPs to import functionalized proteins into plastids for the engineering of RNA processing and metabolic flux. This newly identified cTP has the potential to revolutionize plastid engineering and expand our ability to engineer plastids for biotechnological applications.

## Results

### Identification of 89 polypeptides with differential chloroplast import activities

We initially identified 92 proteins associated with 147 *Arabidopsis* mutants with chloroplast-related phenotypes in the Chloroplast Functional Database (**Fig 1A** and S1 Data) [29,30]. However, four of these proteins were excluded from our study because they were located in organelles other than plastids or required a C-terminal targeting domain for plastid localization (S1 Data). Consequently, we focused on 88 nucleus-encoded proteins (S2 Data). AtRbcS1A cTP, a transit peptide derived from the small subunit of *Arabidopsis* Rubisco (At1g67090), has been widely employed as a model for studying the import of specific proteins into chloroplasts [31]. We used AtRbcS1A cTP as the standard cTP in our analysis. We performed computational analysis with TargetP-2.0 to predict potential cTPs and identify cleavage sites within these proteins (**Fig 1A** and S2 Data) [32]. Of these predicted cTPs, 66 with sequences ranging from 28 to 91 amino acids long from their amino (N)-termini, including cleavage sites, exhibited moderate to high probability scores (cTP scores). These scores predict the likelihood of these sequences being targeted to the stroma (**Fig 1B–1D** and S2 and S3 Data).

Furthermore, our analysis identified 8 transit peptides, each consisting of 68 to 102 residues, that were predicted to be bipartite targeting peptides capable of relocating their preproteins to thylakoids. These peptides are referred to as thylakoid-targeting peptides (tTPs; **Fig 1B–1D and** S2 and S3 Data). By contrast, 15 proteins lacked a probable transit peptide and were predicted to be located in other cellular compartments (**Fig 1B–1D** and S2 and S3 Data). Therefore, we identified a total of 89 polypeptides with differential chloroplast import efficiencies. Our computational prediction did not identify any mitochondrion-targeting peptides (mTPs) within these proteins (S1A Fig, S2 Data and S3 Data). Notably, we observed sequence similarity in the cleavage sites among the predicted cTPs and tTPs, both of which usually contain at least a conserved alanine adjacent to the cleaved amino acids (S1B and S1C Fig and S2 Data).

We studied the chloroplast-targeting activities of the 89 polypeptides using fluorescence tracking. The coding regions of the predicted cTPs and tTPs (or at least the first 60 amino acids of the 15 proteins lacking a predictable transit peptide) were fused to the *GFP(S65T)* gene. We transiently expressed the recombinant proteins in tobacco (*Nicotiana tabacum*) leaves via agroinfiltration. Confocal laser-scanning microscopy (CLSM) and immunoblotting revealed differential expression and subcellular localization of various cTP-GFPs in plant cells (**Fig 1E and 1F**).

Of the 89 fusion proteins, 48 cTP-GFPs showed chloroplast-specific localization (**Figs 1E**, **1F** and S2–S8, and S4 Data). Surprisingly, 8 and 19 cTP-GFPs displayed dual targeting to chloroplasts/mitochondria (Chl/Mt) and chloroplasts/cytosol (Chl/Cy), respectively (**Figs 1E**, **1F** and S2–S8 and S4 Data). Among the 15 cTP-GFPs lacking predicted cTPs, 8 fusion proteins were observed only in the cytosol, and 4 concurrently colocalized to the Chl/Cy or Chl/Mt (S2–S8 Figs and S4 Data). However, 3 other recombinant polypeptide-GFP constructs lacking the predicted transit peptide were observed within chloroplasts (S2–S8 Figs and S4 Data). Fluorescence measurements in the CLSM images demonstrated that the group of chloroplast-specific cTP-GFPs (Chl) exhibited significantly stronger GFP signals within chloroplasts compared to other groups, particularly those with GFP localized to the cytosol (**Figs 1G, 1H** and S9 and S4, S5, and S6 Data). Furthermore, multiple sequence alignment unveiled a conserved cleavage sequence, VR(A/C)(A/S)X, where X represents any amino acid, within 48 cTPs that specifically target GFP transport to chloroplasts (Chl), along with the predicted cTPs demonstrating dual targeting activities (Chl/Mt and Chl/Cy) (S10 Fig).

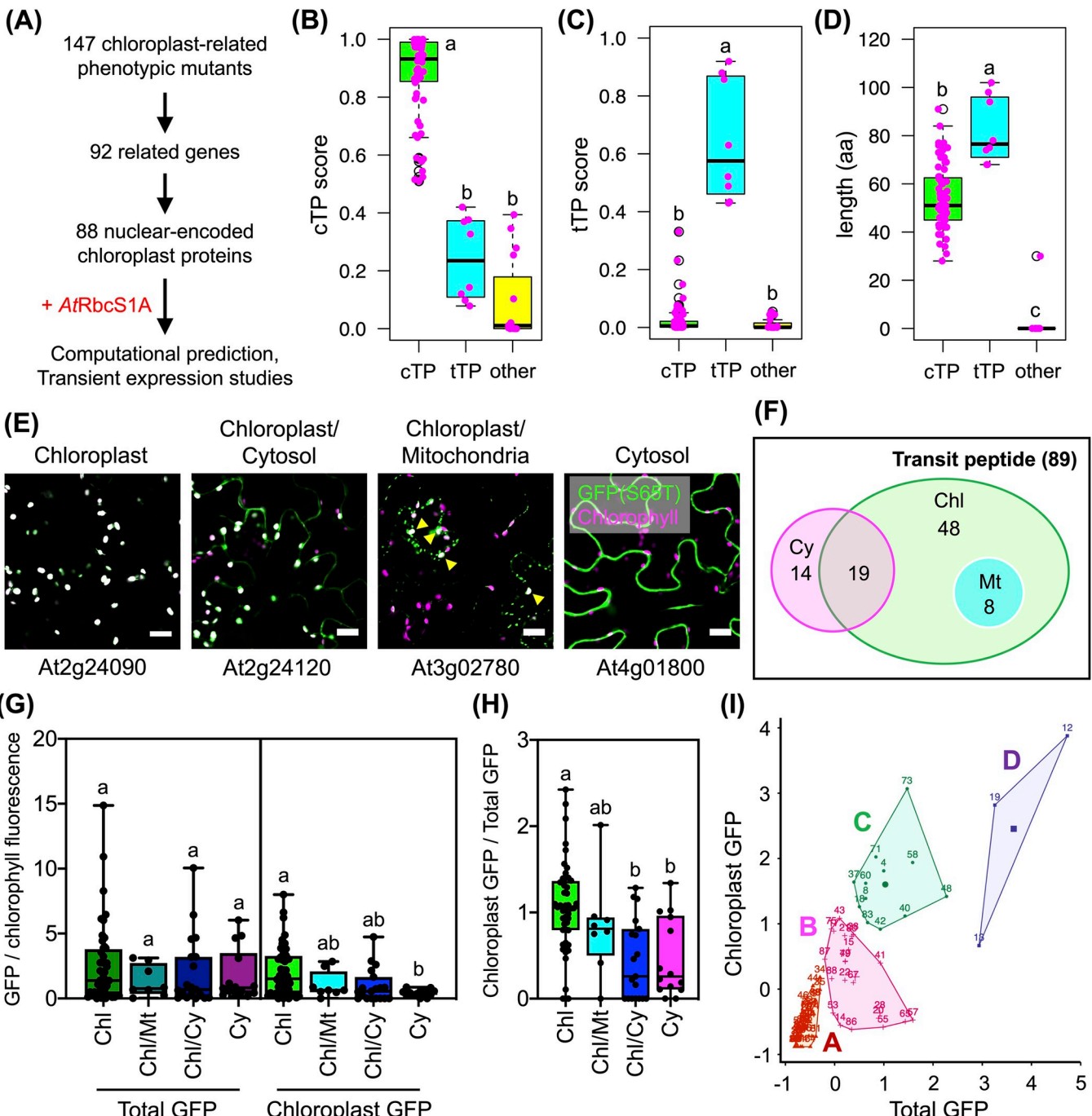

**Fig 1. Screening of high-efficiency chloroplast-targeting peptides.** (**A**) Workflow outlining sequence filtering and collection of plastidial protein sequences from the Chloroplast Mutant Database for computational analysis and identification of putative transit peptide sequences from plastidial proteins. (**B, C**) Computational prediction of cTPs, tTPs, and peptides potentially targeting other organelles in plant cells (referred to as "other") using the N-termini of the 89 plastidial proteins selected for this study. The cTP score and tTP score indicate the likelihood of the predicted peptides being targeted to the chloroplast matrix and thylakoid, respectively. (**D**) Predicted lengths of cTPs, including their possible cleavage sites. Numerical values for **Fig 1B–1D** can be found in S3 Data. (**E**) Subcellular localizations of different cTP-GFPs in tobacco leaf cells. Arrows show chloroplasts. Scale bars = 20 μm. (**F**) Euler diagram of the differential subcellular localizations of 89 cTP-GFPs in plant cells. Chl = Chloroplast, Cy = Cytosol, Mt = Mitochondria. (**G**) Distribution of GFP/chlorophyll fluorescence of recombinant cTP-GFP in plant cells (total GFP) and chloroplasts (chloroplast GFP). (**H**) Ratios of chloroplast GFP to total GFP in fluorescence images of the 4 groups of cTP-GFPs exhibiting different subcellular compartmentalization. Different letters in (**G**) and (**H**) indicate significant differences in the means among the groups of cTP-GFPs, analyzed by one-way ANOVA with Tukey's HSD test at $p = 0.05$. (**I**) Classification of the chloroplast-targeting activities of the cTPs. The 89 cTP-GFPs were categorized into 4 groups based on relative fluorescence intensities in plant cells (Total GFP) vs. inside the chloroplast (Chloroplast GFP) using the $K$-means clustering algorithm. The number of predicted cTPs for each gene locus is listed in S4 and S7 Data. Fluorescence values in **Fig 1G and 1H** are presented in S5 Data. cTP, chloroplast-targeting peptide; GFP, green fluorescent protein; tTP, thylakoid-targeting peptide.

The correlation of the relative intensities of fluorescent signals in plant cells (total GFP) versus inside chloroplasts was analyzed using the *K*-means clustering algorithm. This analysis classified the 89 cTP-GFPs into 4 clusters based on their fluorescence patterns (**Figs 1I** and S11 and S4, S5 and S7 Data). Cluster A consisted of 52 cTP-GFPs that displayed low fluorescence in plant cells, including AtRbcS1A-GFP. The 22 cTP-GFPs in cluster B showed moderate GFP fluorescence in chloroplasts. The 12 cTP-GFPs in cluster C showed strong GFP signals specifically in chloroplasts. The 3 cTP-GFPs in cluster D exhibited intense fluorescence within the chloroplasts and dispersed signals in the cytosol. Together, these findings suggest that various cTPs exhibit differential chloroplast import efficiencies.

We carried out a comparative reassessment of the chloroplast-targeting abilities of different cTPs. Ten representative cTPs that demonstrated noteworthy chloroplast-targeting functions were selected for this analysis (**Fig 2A**). These included 3 from cluster B, 6 from cluster C, and 1 from cluster D. AtRbcS1A cTP, from cluster A, was used as a control (**Fig 2A**). Fluorescence measurements from CLSM images of tobacco leaf cells transfected by *Agrobacterium* harboring different *cTP-GFP* gene expression constructs demonstrated that, despite notable variations, all the selected cTPs exhibited significantly higher chloroplast-targeting efficiencies than AtRbcS1A cTP (**Fig 2B–2D** and S8 and S9 Data). Immunoblot analysis revealed differential expression of the cTP-GFP constructs in plant cells, along with distinct fusion protein levels within chloroplasts (**Fig 2E** and **2F**). Intriguingly, At3g22690-GFP and At4g34740-GFP exhibited mobility-shifted bands in the immunoblots, implying that these proteins did not undergo the appropriate preprotein cleavage following their translocation into chloroplasts (**Figs 2E**, **2F**, and S12). At2g24090-GFP (cluster B) exhibited significantly greater abundance within chloroplasts compared to the other fusion proteins, underscoring the remarkable efficiency of translocation to chloroplasts facilitated by its primary cTP (**Figs 2G** and S12 and S8 and S9 Data).

## The in vitro import abilities of cTPs exhibit negligible differences

We expanded our study to investigate the chloroplast import efficiencies of various cTPs when targeting isolated chloroplasts using in vitro import assays [33]. We primarily focused on At1g63970, At2g20920, and At2g24090 cTPs due to their exceptional chloroplast translocation capabilities, as demonstrated in our in planta expression experiment (**Fig 2**). AtRbcS1A cTP was used as a reference cTP. Our initial approach involved attempting to produce different recombinant cTP-GFPs using the *Escherichia coli* expression system. Unfortunately, this expression platform proved unsuccessful in yielding purified recombinant At2g24090-GFP (**Fig 3A**). It was also difficult to synthesize intact At2g24090-GFP via cell-free synthesis using wheat germ extract, which is consistent with our experiences with the *E. coli* expression system (**Fig 3B**). Comprehensive immunoblot blot analysis revealed unexpected cleavage of recombinant At2g24090-GFP by bacterial peptidases following its induced expression in *E. coli* cells (**Fig 3C** and **3D**). The nonspecific cleavage of intact At2g24090-GFP by both bacterial and plant peptidases suggests the importance of the predictable cleavage site in determining the specificity and varying import efficiencies of the cTPs.

Despite the difficulties encountered in synthesizing intact recombinant proteins, we utilized successfully purified recombinant cTP-GFPs, including AtRbcS1A-GFP, At1g63970-GFP, and At2g20920-GFP, along with purified fractions of At2g24090-GFP, to further investigate their import into isolated chloroplasts. As a negative control in the in vitro import assay, we also included recombinant GFP lacking the cTP portion (free GFP). We analyzed the import efficiencies of different recombinant cTP-GFPs into isolated tobacco chloroplasts using time-course in vitro import assays, along with immunoblotting of proteins recovered from the

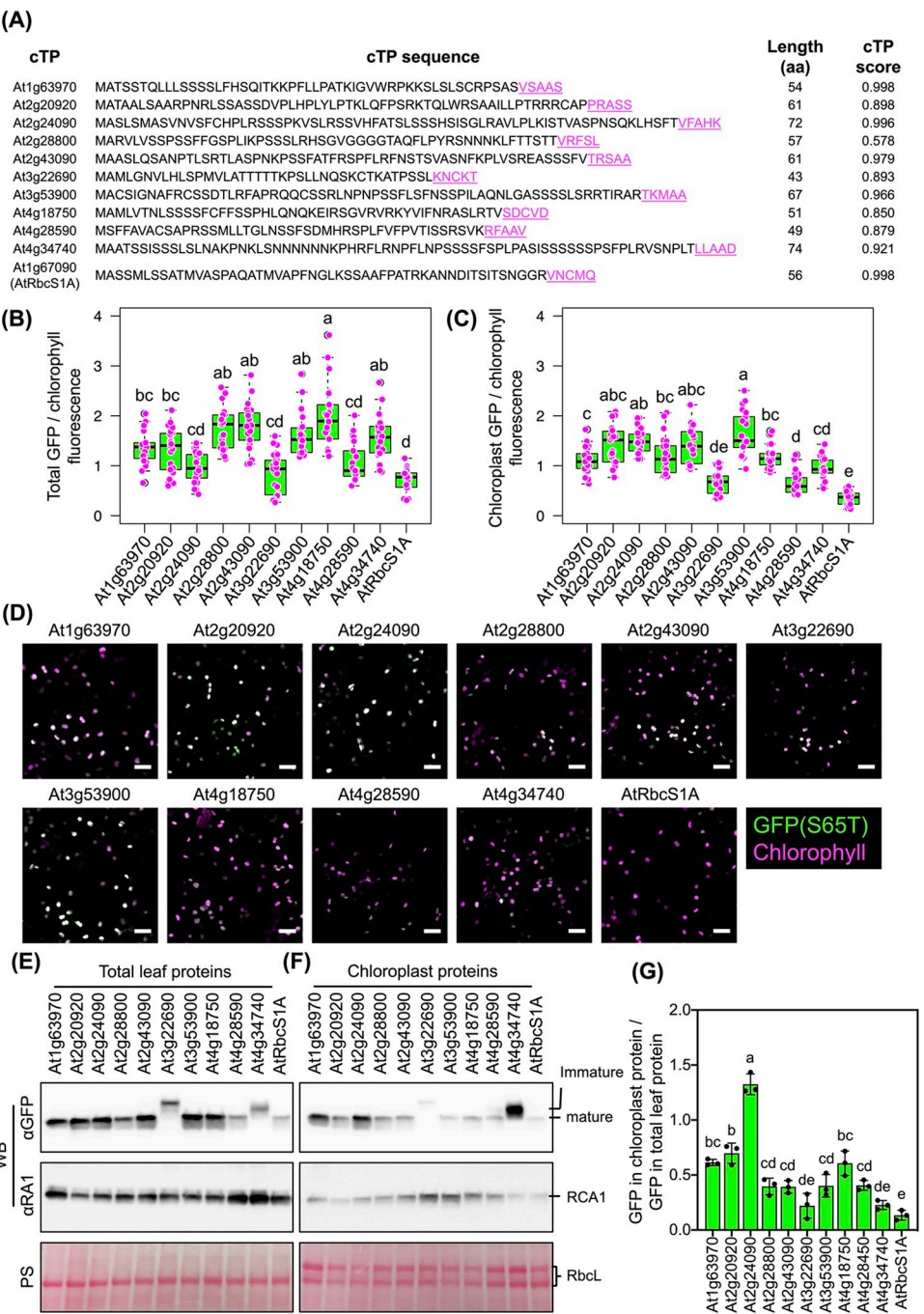

**Fig 2. Comparative analysis of selected cTPs.** (**A**) Amino acid sequences of selected cTPs whose import efficiencies into chloroplasts were re-assessed. The underlined amino acid sequences shown in magenta are the predicted cleavage sites within each cTP. The cTP score reports the predicted probability that these transit peptides are targeted to chloroplasts. (**B, C**) Distribution of GFP/chlorophyll fluorescence in plant cells and chloroplasts, presented as box plots ($n$ = 21, see numerical data in S9 Data). The black bars within the box plots represent the medians of the data distribution. Different letters indicate significant differences in the means, determined using one-way ANOVA with Tukey's HSD test at $p$ = 0.05. (**D**) CLSM images of tobacco leaf cells expressing different cTP-GFPs. Scale bars = 20 μm. (**E, F**) Immunoblotting of cTP-GFP in total leaf proteins and isolated chloroplast proteins, respectively. PS = Ponceau S staining, αRA1 = anti-Rubisco Activase 1 antibody, αGFP = anti-GFP antibody, RbcL = Rubisco-large subunit, and RCA1 = Rubisco activase 1. (**G**) Quantitative analysis of GFP levels in chloroplasts per in total leaf proteins. Error bars represent SD ($n$ = 3, S9 Data). Different letters indicate significant differences in the means, analyzed using one-way ANOVA with Tukey's HSD test at $p$ = 0.05. CLSM, confocal laser-scanning microscopy; cTP, chloroplast-targeting peptide; GFP, green fluorescent protein; SD, standard deviation.

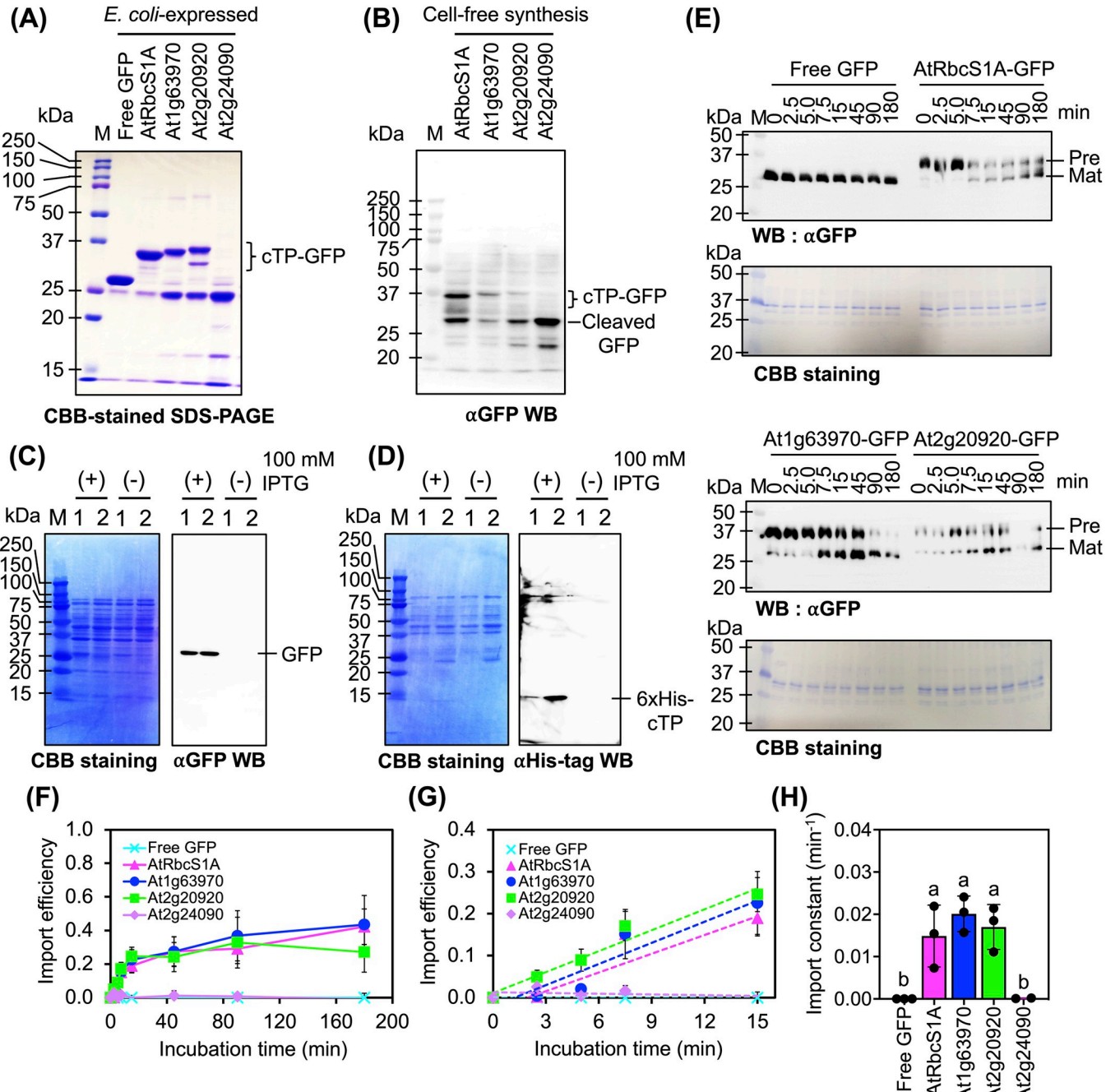

**Fig 3. In vitro import assays of recombinant cTP-GFPs.** (**A**) SDS-PAGE of various recombinant cTP-GFPs prepared using the *E. coli* expression system. One microgram of recombinant proteins was analyzed on a 14% SDS-PAGE gel before staining with CBB to visualize the protein bands. (**B**) Immunoblot analysis of recombinant cTP-GFPs expressed in cell-free protein synthesis reactions. Two microliters of each cell-free synthesis reaction was resolved by SDS-PAGE and immunoblotted using anti-GFP antibody. (**C, D**) Cleavage of recombinant At2g24090 cTP-GFP in *E. coli* cells. Cell lysates extracted from *E. coli* cultures with (+) and without (-) induction by IPTG were subjected to immunoblotting using anti-GFP (right panel in (**C**)) and anti-His tag (right panel in (**D**)) antibodies. The membranes were stained with CBB after immunoblotting (left panels in (**C, D**)). (**E**) Immunoblot analysis of proteins with in vitro import activity. The abundance of different cTP-GFP precursors (Pre) and mature proteins (Mat) in import reactions collected at various time points post-incubation was analyzed by immunoblotting with anti-GFP antibody. The membranes were stained with CBB after immunoblotting to ensure equal loading of protein samples onto the membrane. (**F, G**) Import efficiencies of various recombinant cTP-GFPs into chloroplasts. The efficiencies of 3 selected cTPs for importing GFP into chloroplasts were determined via in vitro import assays as shown in (**E**). (**G**) Shows the linear regression of the import efficiencies of different cTP-GFPs at 0 to 15 min post-incubation. (**H**) Import constants of each cTP-GFP. Import constants for the recombinant cTP-GFPs were calculated using the linear regression equations of each protein at 2.5 to 15 min of incubation. Error bars in (**F–H**) represent the standard deviations of the mean from 3 biologically independent experiments ($n = 3$). Different letters indicate significant differences among the recombinant proteins (one-way ANOVA with Tukey's HSD test at $p = 0.05$). Quantitative data for **Fig 3F–3H** can be found in S10 Data. CBB, Coomassie brilliant blue; cTP, chloroplast-targeting peptide; GFP, green fluorescent protein.

import reactions (**Figs 3E** and S13). N-terminal cTPs are typically cleaved by stromal processing peptidase during and after the translocation of preprotein into the chloroplast [6,34]. Consequently, we refer to the unimported form of cTP-GFP as the "precursor" and the imported form as the "mature" form.

Unlike free GFP, whose appearance did not change, 3 proteins, AtRbcS1A-GFP, At1g63970-GFP, and At2g20920-GFP, were imported into isolated tobacco chloroplasts. In all cases, cTP-GFP import and the maturation of imported proteins were time dependent (**Figs 3E** and S13). However, we did not detect any residue of At2g24090-GFP in the import reactions (S13 Fig). Surprisingly, the import efficiencies and import rates of AtRbcS1A, At1g63970, and At2g20920 cTPs were not significantly different (**Fig 3F–3H** and S10 Data). These findings suggest that cTPs may require an additional import mechanism or other scaffold proteins to facilitate efficient translocation of proteins into chloroplasts.

## In vivo import efficiencies of the selected cTPs

Targeting a chloroplast transit peptide (cTP)-fused fluorescent protein serves as a pivotal marker, with its abundance within the designated organelle being regulated through distinct mechanisms, including mRNA expression, protein translation, stability, and proper folding and formation of the fluorophores, both within the broader cellular context and specifically within chloroplasts. These mechanisms collectively influence the efficiency with which various recombinant cTP-GFPs are imported into chloroplasts. To explore the dynamics of chloroplast import, we conducted a time-course expression analysis focusing on specific cTPs (At1g63970, At2g20920, and At2g24090), using AtRbcS1A as a reference cTP. This involved the agroinfiltration of expression cassettes containing the selected cTP-GFPs, alongside controls such as free GFP for fluorescence comparison and a negative control vector (pBI121-EV), into tobacco leaves. Subsequent leaf sampling at various time points post-agroinfiltration (24, 36, 48, 60, 72, and 96 h) enabled comprehensive expression profiling and in vivo import assessments.

Quantitative reverse-transcription PCR (qRT–PCR) and fluorometric measurements were employed to evaluate transcript expression and accumulation of recombinant cTP-GFPs in transformed plant leaves, respectively. Our transcript expression analysis unveiled peak expression levels of all *cTP-GFP* transcripts in agroinfiltrated plant leaves at 36 hours after infiltration (HAI), followed by a gradual decline from 48 to 96 HAI (**Fig 4A** and S11 Data). Interestingly, significant variations in transcript levels were observed among samples transformed with different *cTP-GFP* expression vectors at these time points (**Fig 4A** and S11 Data). Fluorescence measurement revealed higher accumulation of fluorescent proteins in total leaf proteins extracted from leaves of plants transformed with cTP-GFPs and GFP-expression vectors compared to the EV control at 36 to 96 HAI (**Fig 4B** and S11 Data). However, no statistically significant differences in accumulation of fluorescent proteins were detected among the leaves of plants transformed with different cTP-GFP constructs (**Fig 4B** and S11 Data). Unexpectedly, despite constitutive expression of various *cTP-GFP* genes, higher transcript levels did not translate into significantly greater accumulation of recombinant cTP-GFP in transformed plant leaves (**Fig 4A** and **4B** and S11 Data).

The localization of recombinant cTP-GFPs within chloroplasts was investigated using CLSM imaging, followed by quantification of GFP fluorescence both in plant cells and within chloroplasts. Surprisingly, AtRbcS1A-GFP was found to partially localize in the cytosol from 24 to 60 HAI, gradually transitioning to chloroplast targeting from 72 to 96 HAI (**Figs 4C, 4D** and S14 and S11 Data). In contrast, recombinant At1g63970-, At2g20920-, and At2g24090-GFP specifically localized within chloroplasts from 24 to 96 HAI (**Figs 4C, 4D** and S14 and S11 Data). Remarkably, the fluorescent intensities of chloroplast-localized At1g63970-, At2g20920-, and

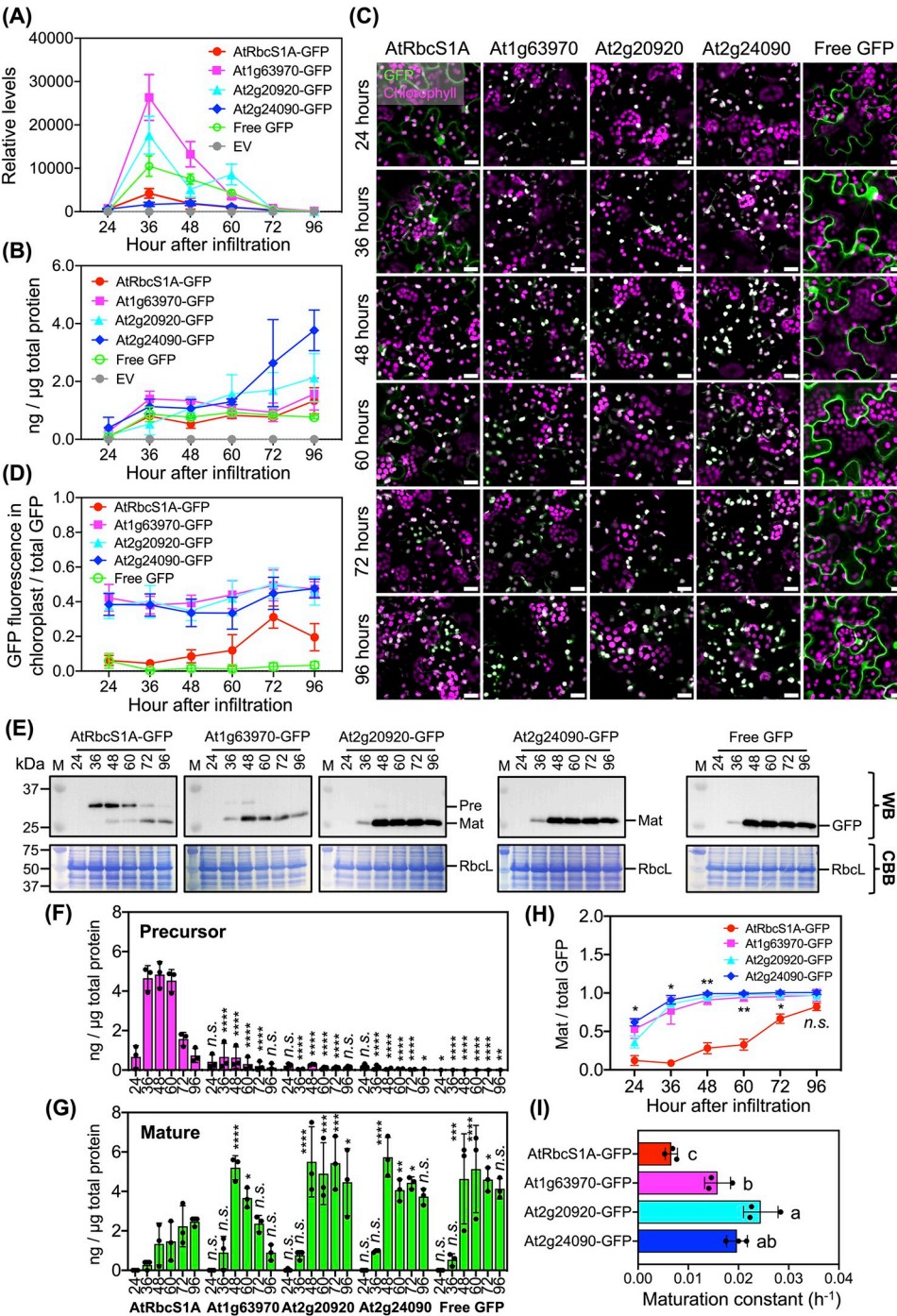

**Fig 4. In vivo translocation of cTP-GFPs.** (**A**) Transcript expression of *cTP-GFP* genes in agroinfiltrated tobacco leaves at different time points post-agroinfiltration analyzed by qRT-PCR using *GFP*-specific primers. The expression levels of *cTP-GFP* transcripts were compared to that of pBI121-transformed leaves (EV) at 24 h post-infiltration. Relative transcript expression was determined from 3 biologically independent agroinfiltrated leaves (*n* = 3, S11 Data). Error bars represent SD. (**B**) Abundance of various cTP-GFP fluorescent proteins in total leaf proteins. GFP fluorescence in total proteins extracted from agroinfiltrated tobacco leaves at different time points was measured, and the amounts of fluorescent protein in 3 experimentally independent samples (*n* = 3, S11 Data) were calculated from the linear regression equations generated from the fluorescent intensities of standard GFP. Error bars represent SD. (**C**) Localization of various recombinant cTP-GFPs in transfected tobacco leaf cells at different time points post-agroinfiltration. Scale bars: 20 μm. (**D**) Ratio of GFP fluorescence in chloroplasts to total GFP fluorescence in a CLSM image. CLSM images were captured from transformed plant leaves at different time points. Fluorescent intensities of

recombinant cTP-GFPs within chloroplasts and in an ROI of a CSLM image were measured using Fiji ImageJ. Error bars represent SD of 16 CLSM images (*n* = 16, S11 Data) collected from 3 independent experiments. (**E**) Immunoblot analysis of recombinant cTP-GFPs in total leaf proteins. Total proteins from agroinfiltrated tobacco leaves at different time points post-infiltration were immunoblotted (WB) against anti-GFP polyclonal antibody to detect the recombinant cTP-GFP precursors (Pre) and the chloroplast-localized matured proteins (Mat) in samples. Equal loading of proteins onto the immunoblot membrane was confirmed by CBB staining of the Rubisco large subunit protein (RbcL) band. (**F, G**) Accumulation of various cTP-GFP precursors (**F**) and their matured proteins after translocation to chloroplasts (**G**). The content of recombinant proteins in total leaf proteins was determined from 3 biologically independent immunoblot experiments against the calibration equation of different amounts of standard GFP (*n* = 3, S11 Data). Error bars represent SD. (**H**) Increase in matured GFP contents in total leaf proteins from 24 to 96 HAI. Error bars represent SD of the means of matured protein contents in total leaf proteins from 3 biologically independent samples (*n* = 3, S11 Data). Asterisks in (**F**) to (**H**) indicate statistically significant differences analyzed by ANOVA with Dunnett's multiple comparisons against the AtRbcS1A-GFP contents (*; $P \leq 0.05$, **; $P \leq 0.01$, ***; $P \leq 0.001$, and ****; $P < 0.0001$). n.s. represent no significant difference. (**I**) Maturation rate constants of different cTP-GFP. Rate constants were determined from linear regression equations of matured protein contents in total leaf proteins at 24 to 48 HAI. Error bars represent SD of the average of rate constants from 3 experiments (*n* = 3, S11 Data). Letters in (**I**) indicate different levels of statistically significant difference of means analyzed by one-way ANOVA with Tukey's HSD test at *p* = 0.05. CBB, Coomassie brilliant blue; CLSM, confocal laser-scanning microscopy; cTP, chloroplast-targeting peptide; EV, empty vector; GFP, green fluorescent protein; HAI, hours after infiltration; qRT-PCR, quantitative reverse-transcription PCR; ROI, regions of interest; SD, standard deviation.

At2g24090-GFPs were higher than those of AtRbcS1A-GFP at all time points post-agroinfiltration (**Figs 4D** and S14 and S11 Data).

Furthermore, the chloroplast import activities of different recombinant cTP-GFPs were investigated using immunoblot analysis. Total leaf proteins were extracted from transformed leaf samples collected at 24 to 96 HAI and immunoblotted against an anti-GFP antibody. The immunoblotting results revealed differential accumulations of precursor cTP-GFPs and their mature chloroplast-localized forms in total leaf proteins (**Figs 4E** and S15 and S11 Data). A gradual decrease in AtRbcS1A-GFP precursors and an increase in its mature proteins were detected over 24 to 96 HAI in transformed leaves (**Figs 4E–4G** and S15 and S11 Data). Although cTP-GFP precursors of At1g63970- and At2g20920-GFPs were also detected up to 48 HAI, their contents in total leaf proteins were significantly lower than that of AtRbcS1A-GFP at the same time points (**Figs 4E, 4F** and S15 and S11 Data). However, no detectable At2g24090-GFP precursor was observed on the immunoblot membrane (**Figs 4E, 4F** and S15 and S11 Data). Additionally, quantitative immunoblot analysis revealed significantly higher accumulations of mature At1g63970-, At2g20920-, and At2g24090-GFPs in total leaf proteins compared to mature AtRbcS1A-GFP (**Fig 4G** and S11 Data). The accumulations of these 3 mature cTP-GFPs in total leaf proteins progressively increased up to 48 HAI, with a significantly higher maturation rate than AtRbcS1A-GFP (**Fig 4H and 4I** and S11 Data). However, unlike mature At2g20920- and At2g24090-GFPs, which showed no degradation after 48 HAI, mature At1g63970-GFP gradually degraded over the subsequent period post-infiltration (**Figs 4E, 4G** and S15 and S11 Data). Our CLSM imaging and immunoblot analysis suggest that, despite differential degradation of mature variants, the 3 selected cTPs have higher import efficiencies in transporting GFP into chloroplasts in transformed plant cells compared to the reference AtRbcS1A cTP.

## The cleavage site is essential for the strong chloroplast import ability of At2g24090 cTP

The length of a specific cTP plays a pivotal role in facilitating the efficient translocation of proteins into chloroplasts [7,14,18,35]. The current findings underscore the exceptional import activities of the selected cTPs including At1g63970, At2g20920, and At2g24090 in delivering conjugated GFP to chloroplasts in tobacco leaf cells, surpassing that of other scanned cTPs.

The complimentary amino acid sequence, typically associated with the proximal length of the cTP, interacts with the chloroplast import machinery such as proteins responsible for scaffold formation and quality control in the cytosol. Notably, At2g24090-GFP exhibited an unexpected susceptibility to both bacterial and plant peptidases (**Fig 3A**–**3D**). Given potential factors including the specificity of the cTP sequence to peptidases and the 3D structure of cTPs, it is theoretically possible that At2g24090 cTP could be cleaved by other peptidases within a random sequence context rather than at its predicted cleavage site, thereby facilitating the targeting of conjugated proteins into chloroplasts. Consequently, further investigations into the effective length of At2g24090 cTP and the potential involvement of its cleavage site in mediating efficient protein translocation to the chloroplast are warranted.

In our initial approach, we designed several recombinant GFP-fused C-terminal-depletion mutants of At2g24090 cTP (ranging from 15 to 65 amino acids from its N-terminus) for in vitro import studies (**Figs 5A** and S16A). Notably, unlike full-length At2g24090-GFP, all of the truncated At2g24090 cTP-GFPs exhibited increased stability, and we successfully purified these recombinant proteins from *E. coli* cells (S16A Fig). In vitro import analysis at 3 h after ATP addition revealed that all of the truncated At2g24090 cTPs exhibited slightly different abilities to transport GFP to isolated tobacco chloroplasts (**Figs 5B**, **5C** and S16B and S12 Data). However, a time-course in vitro import study unveiled distinct import efficiencies correlated with the varying lengths of At2g24090 cTPs (**Fig 5D**–**5F** and S12 Data). Truncated At2g24090 cTPs shorter than 35 amino acids from the N-terminus did not enable time-dependent import (**Fig 5D** and **5E** and S12 Data). However, import efficiencies significantly improved as the length of the amino acid sequences in the truncated cTPs increased (**Fig 5E** and **5F** and S12 Data). Based on this in vitro import study, the optimal length of At2g24090 cTP for effective translocation of the conjugated protein into the chloroplasts appears to be within 1 to 55 amino acids from the N-terminus.

We conducted an extensive investigation to assess the import efficiencies of various truncated At2g24090 cTP-GFPs into chloroplasts within plant cells using in planta expression and fluorescence tracking. To achieve this, we subcloned the coding sequences of recombinant truncated At2g24090 cTP-GFPs (as shown in **Fig 5A**) into plant expression vectors and introduced these vectors into tobacco leaf cells by agroinfiltration. CLSM imaging revealed distinct subcellular localizations of the recombinant fluorescent proteins within plant cells (**Figs 5G** and S17A). Specifically, GFP-fused truncated At2g24090 cTPs containing 15 and 25 amino acids from their N-termini (referred to as 15-GFP and 25-GFP) localized to the cytosol of transfected leaf cells. By contrast, recombinant 35-GFP and 45-GFP exhibited a dual distribution pattern in both the cytosol and chloroplasts of plant cells. Remarkably, we observed chloroplast-specific targeting of recombinant cTP-GFPs in transformed plant cells when longer truncated At2g24090 cTPs (55 and 65 amino acids; designated as 55-GFP and 65-GFP, respectively) were used. Notably, however, the chloroplast-specific fluorescent signals of these recombinant proteins were lower in intensity compared to those of full-length At2g24090-GFP (FL-GFP; **Figs 5G** and S17A).

To further test the chloroplast-specific translocation of these truncated variants of At2g24090 cTP-GFPs within plant cells, we performed immunoblot analysis of both total leaf proteins and chloroplast proteins collected from agroinfiltrated tobacco leaves. Consistent with our CLSM imaging results, the recombinant cTP-GFP constructs encompassing truncated At2g24090 cTPs ranging from 15 to 45 amino acids from the N-terminus did not exhibit robust accumulation within the chloroplasts (**Figs 5H**, **5I**, and S17B and S12 Data). By contrast, recombinant 55-GFP and 65-GFP displayed increased abundance in chloroplast proteins, indicating that they were successfully targeted to chloroplasts (**Figs 5H**, **5I**, and S17B and S12 Data). Nevertheless, 55-GFP and 65-GFP accumulated to significantly lower levels within

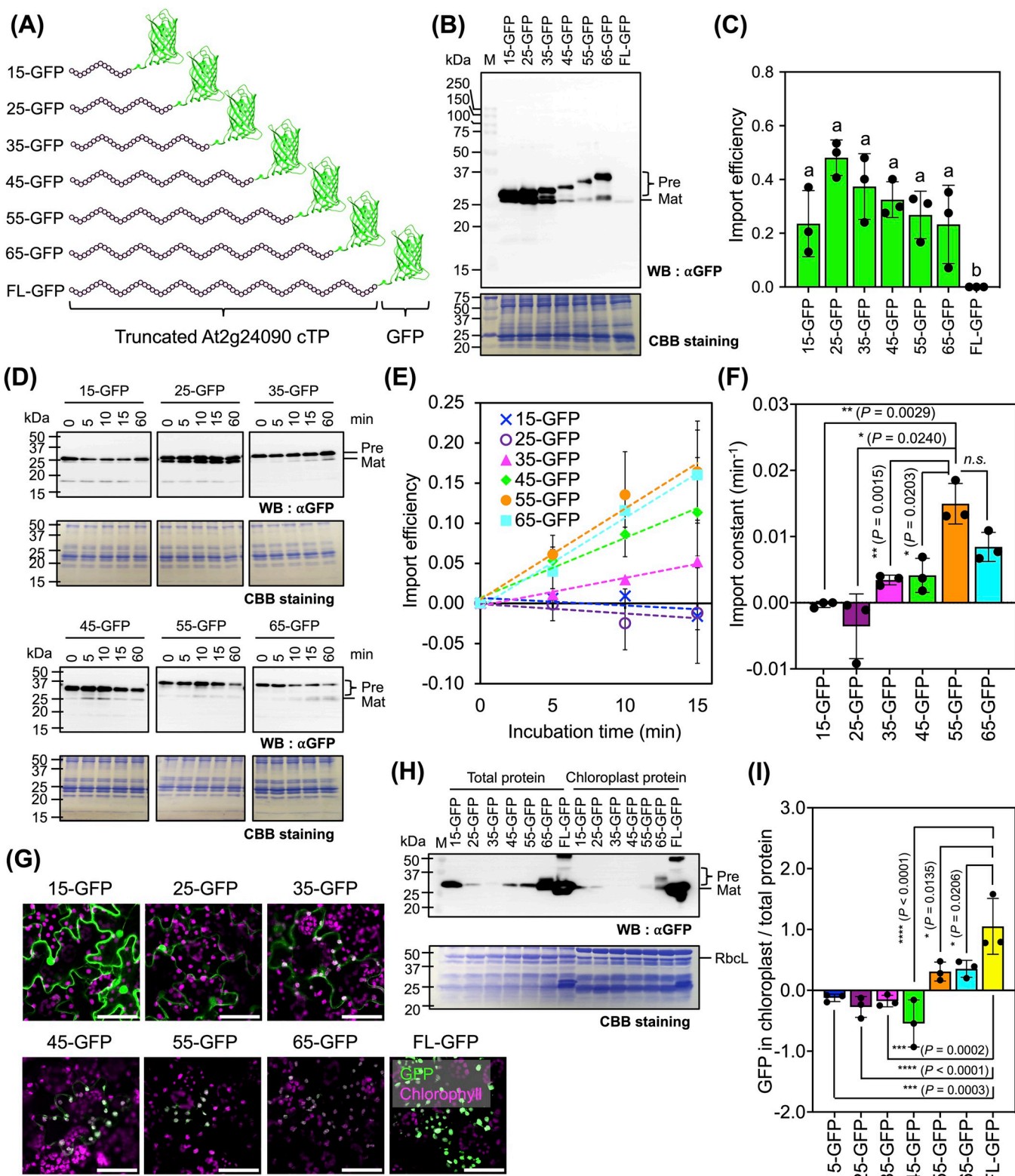

**Fig 5. The cleavage site is crucial for the efficient translocation of At2g24090 cTP.** (**A**) Diagram illustrating recombinant truncated At2g24090 cTP-GFPs. (**B**) Immunoblot analysis of proteins recovered at 3 h after the import reaction of truncated At2g24090-GFPs and isolated tobacco chloroplasts using anti-GFP antibody. CBB staining shows equal protein loading. "Pre" indicates the precursor forms of recombinant At2g24090-GFPs; "Mat" represents matured (cleaved) cTP-GFPs. (**C**) Import efficiencies of At2g24090-GFP variants determined by immunoblotting as in (**B**). Error bars represent standard deviations of the mean

from 3 independent experiments ($n$ = 3, S12 Data). Different letters indicate significant differences among treatments (one-way ANOVA with Tukey's HSD test at $p$ = 0.05). (**D**) Time-dependent import functions of truncated At2g24090-GFPs. Proteins extracted from import reactions at different time points (0–15 min) were analyzed by immunoblotting using anti-GFP antibody. CBB staining shows equal protein loading. (**E**) Import efficiencies of truncated At2g24090-GFPs. Chloroplast import efficiencies of recombinant proteins at different time points were determined by immunoblotting (as shown in (**D**)). Error bars represent standard deviations of the mean from 3 assays ($n$ = 3, S12 Data). (**F**) Import constants of truncated At2g24090-GFPs into chloroplasts, calculated from the linear regression in (**E**) at time = 0 to 15 min (see numerical values in S12 Data). Error bars represent standard deviations. Asterisks indicate significant differences in the means of each treatment compared to 55-GFP (Student's $t$ test: *; $p$ < 0.05, **; $p$ < 0.01). n.s., no significant difference. (**G**) Subcellular localization of At2g24090-GFP variants in agroinfiltrated tobacco leaf cells at 3 DAI. Scale bars are 50 μm. (**H**) Immunoblot analysis of recombinant cTP-GFPs in total leaf proteins and isolated chloroplast proteins at 3 DAI using anti-GFP antibody to detect recombinant cTP-GFP levels in plant proteins prior to CBB staining. RbcL, large subunit of Rubisco. (**I**) Quantitative analysis of the import efficiencies of truncated At2g24090-GFPs in chloroplasts after agroinfiltration. The import efficiency of each cTP-GFP was determined by immunoblotting (**H**). Error bars represent standard deviations of the mean from 3 independent experiments ($n$ = 3, S12 Data). Asterisks indicate significant differences (Student's $t$ test: *; $p$ < 0.05, **; $p$ < 0.01, n.s., no significant difference). CBB, Coomassie brilliant blue; cTP, chloroplast-targeting peptide; GFP, green fluorescent protein.

chloroplasts than the more efficiently targeted FL-GFP (**Figs 5H, 5I**, and S17B and S12 Data). Collectively, the results of in vitro import and in planta expression analyses indicate that the efficient translocation of plastidial proteins requires the full-length At2g24090 cTP, with particular emphasis on its cleavage site.

## At2g24090 cTP is a highly efficient transit peptide for metabolic engineering

Exchanging the original transit peptide with a highly effective cTP improved the organellar functions of chloroplast-specific proteins [5]. *Arabidopsis* AtGGPPS2 and AtPSY1 are nucleus-encoded plastidial enzymes that control metabolic steps in the methylerythritol 4-phosphate (MEP) pathway leading to the biosynthesis of chlorophylls and carotenoids (**Fig 6A**) [36]. We hypothesized that replacing the transit peptides in AtGGPPS2 and AtPSY1 with a highly efficient one, such as At2g24090 cTP, would enhance the targeting of recombinant enzymes to chloroplasts and facilitate the effective engineering of chlorophyll and carotenoid biosynthesis within chloroplasts. Based on this hypothesis, we performed a comparative analysis to verify the better chloroplast-targeting activity of At2g24090 cTP over the native cTP portions of AtGGPPS2 and AtPSY1. We constructed T-DNA vectors harboring the expression cassettes of recombinant AtGGPPS2 cTP- and AtPSY1 cTP fused to GFP(S65T) for in planta expression and fluorescence imaging (S18A Fig). Subsequently, tobacco leaves were transformed with these constructs and the At2g24090-GFP expression vector via agroinfiltration. CLSM imaging was carried out to validate the different efficiencies of these cTPs to translocate GFP into chloroplasts. Fluorescent measurement of CLSM images and immunoblotting indicated that At2g24090 cTP had a significantly greater ability than AtGGPPS2 and AtPSY1 cTPs to translocate GFP into chloroplasts, especially AtPSY1 cTP (S18B–S18D Fig and S13 Data).

To further validate the enhanced import efficiency of recombinant enzymes into chloroplasts, we constructed plant expression vectors for the expression of recombinant C-terminal FLAG (DYKDDDDK)-tagged AtGGPPS2 (AtGGPPS2-FLAG). We then replaced the cTP portion of the AtGGPPS2 enzyme with the full-length At2g24909 cTP, creating a recombinant At2g24090 cTP-GGPPS2-FLAG enzyme (S19A Fig). We investigated the translocation of these recombinant enzymes into chloroplasts through immunoblot analysis of total leaf proteins and chloroplast proteins isolated from agroinfiltrated tobacco leaf cells. At2g24090 cTP exhibited a significantly enhanced ability to translocate the recombinant enzymes into chloroplasts (S19B and S19C Fig and S14 Data).

Effectively modulating metabolic processes within specific plastids is crucial for successful metabolic engineering. In pursuit of this goal, we replaced the cTP domains of AtGGPPS2 and AtPSY1 with At2g24090 cTP (**Fig 6B**). We introduced these modified expression cassettes into

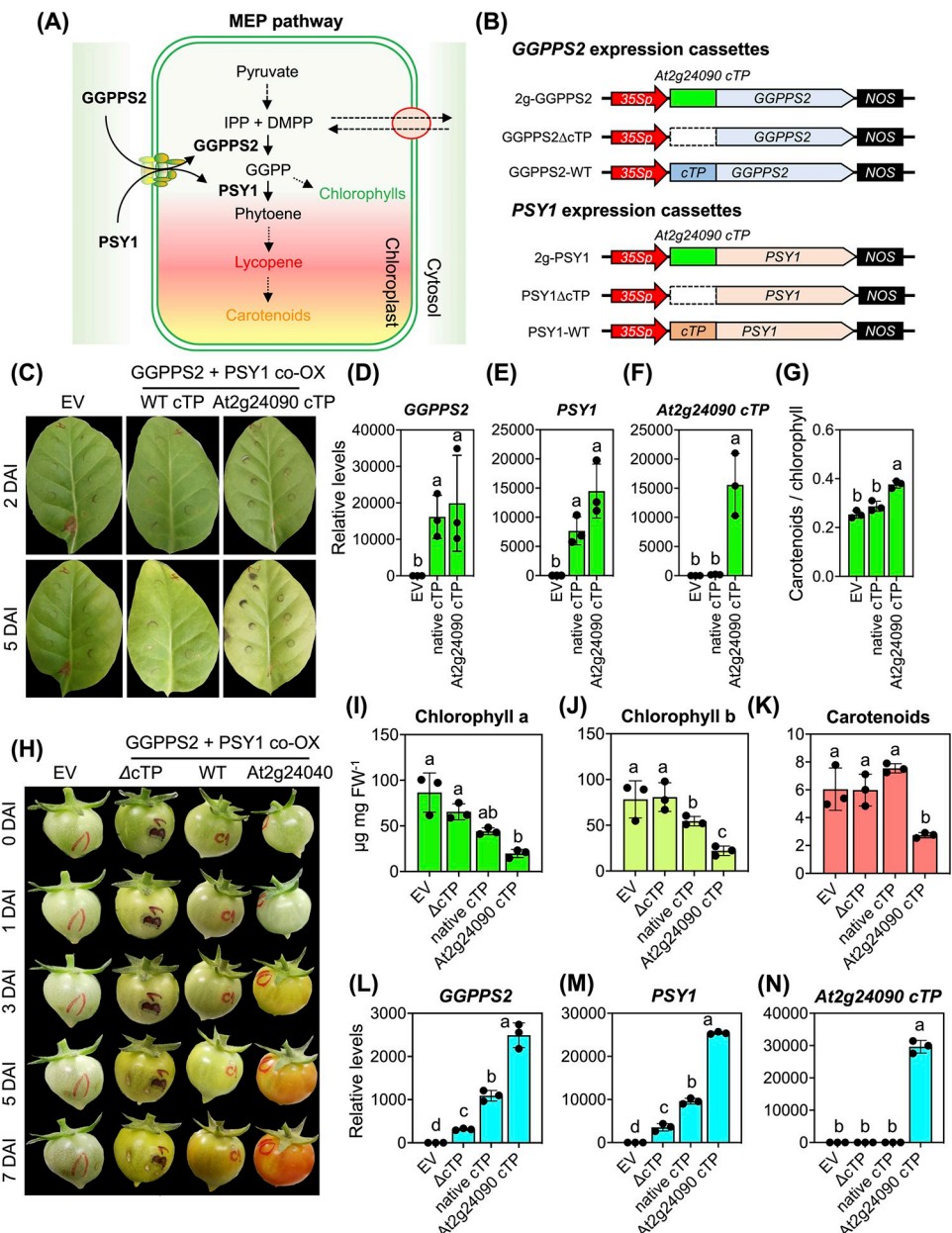

**Fig 6. Expression of cTP-engineered chloroplast enzymes alters the metabolic traits of plants.** (**A**) Involvement of GGPPS2 and PSY1 in the plastidial MEP pathway. (**B**) Expression vectors of cTP-engineered enzymes. (**C**) Changes in leaf color at 2 and 5 DAI. WT cTP = tobacco leaves coexpressing native enzymes (native cTP). At2g24090 cTP = leaves coexpressing At2g24090 cTP-engineered enzymes. (**D–F**) Transcript levels of enzyme-encoding genes and *At2g24090 cTP* mRNA in tobacco leaves at 5 DAI. Relative transcript expression values are presented in S16 Data. (**G**) Ratio of carotenoids to total chlorophyll contents in tobacco leaf cell exudates at 5 DAI (see values in S16 Data). (**H**) Acceleration of tomato fruit ripening after coinjection of cTP-engineered enzyme expression vectors. (**I–K**) Pigment contents of agroinjected tomato fruits at 7 DAI (see data in S16 Data). (**L–N**) Transcript levels in tomato fruits at 7 DAI. Error bars are standard deviations of the mean from 3 independent experiments (*n* = 3, S16 Data). Different letters indicate significant differences in the means (one-way ANOVA with Tukey's HSD test at *p* = 0.05). cTP, chloroplast-targeting peptide; DAI, days after agroinfiltration; MEP, methylerythritol 4-phosphate.

both tobacco leaves and tomato fruits using *Agrobacterium*-mediated gene transfer. Notably, individual expression of the cTP-engineered enzymes did not induce discernible changes in the metabolic traits of tobacco leaves, suggesting that significant alterations in related

metabolic processes require a combination of multiple factors (S20A and S20B Fig and S15 Data). Coexpression of the native AtGGPPS2 and AtPSY1 enzymes in tobacco leaves, as compared to the control (EV control), had no significant effect on pigment contents (**Figs 6C–6G** and S20 and S15 and S16 Data). By contrast, the dual expression of At2g24090 cTP-engineered enzymes led to a notable reduction in chlorophyll a and b levels in chlorotic tobacco leaves at 5 days post-infiltration (**Figs 6C–6G** and S20 and S15 and S16 Data). This coexpression also resulted in reduced chlorophyll and carotenoid levels while accelerating the maturation of transfected tomato fruits (**Fig 6H–6N** and S16 Data). These findings collectively suggest that At2g24090 cTP serves as a highly efficient transit peptide for precisely targeting functional enzymes to plastids, underscoring its potential importance for metabolic engineering.

## A cTP-engineered RNA-processing protein can enhance plastidial RNA stability

Photosystems are essential complexes responsible for oxygenic photosynthesis that are situated on the thylakoid membranes within chloroplasts (**Fig 7A**). Numerous proteins in these functional centers, including those within electron transport complexes (e.g., the cytochrome *b6f* complex), are encoded by photosynthesis-related genes, which are organized in the plastid *psbB* operon (**Fig 7A** and **7B**). The plant HCF152 RNA-binding protein is a nucleus-encoded plastid-targeted pentatricopeptide repeat protein (PPR) that plays a pivotal role in the processing of transcripts associated with photosynthesis, especially those within the *psbB* operon (**Fig 7B**) [37,38]. Mutations in Arabidopsis *AtHCF152* disrupt the RNA processing of the *psbB* operon, leading to a reduction in the photosynthetic function of chloroplasts [38]. Similarly, suppressing *HCF152* gene expression in *N. benthamiana* through virus-induced gene silencing (VIGS) resulted in the disruption of intercistronic RNA processing and a decrease in the abundance of *petB* transcripts within the plastidial *psbB* operon [39]. Conversely, we reasoned that selectively targeting tobacco NtHCF152 into chloroplasts using the highly efficient At2g24090 cTP could potentially enhance the abundance of transcripts within the plastidial *psbB* operon in transformed tobacco cells, thereby increasing the photosynthetic capacity of the resulting engineered plants.

To explore this notion, we initially assessed the import efficiency of NtHCF152 cTP compared to the highly efficient At2g24090 cTP using in planta expression and fluorescence tracking techniques. We introduced recombinant fluorescent cTP-GFP expression vectors (**Fig 7C**) into tobacco leaf cells via agroinfiltration. Our analyses, involving fluorescence measurements and immunoblotting of cTP-GFPs in tobacco leaves, demonstrated that At2g24090 cTP exhibited significantly greater efficacy than NtHCF152 cTP in translocating GFP to chloroplasts (**Fig 7D–7G** and S17 Data).

Subsequently, we modified NtHCF152 cTP by eliminating its cTP domain or substituting it with At2g24090 cTP (**Fig 7H**). We introduced the resulting expression constructs (**Fig 7H**) into tobacco leaves via agroinfiltration and measured transcript levels by qRT-PCR. We observed higher levels of *NtHCF152* and *At2g24090 cTP* transcripts in leaves infiltrated with the At2g24090 cTP-HCF152 expression vector versus the control (**Fig 7I** and **7J** and S17 Data). We detected significant increases in *psbT*, *petB*, and *petD* transcript levels in leaves expressing At2g24090 cTP-HCF152 compared to those expressing native NtHCF152 protein (**Fig 7K** and S17 Data). However, there were no discernible changes in the levels of other transcripts within the same polycistronic RNA, including *psbB*, *psbN*, and *psbH* (**Fig 7K** and S17 Data). These results point to the presence of other RNA-binding proteins that likely regulate the abundance of these transcripts. Importantly, no differential expression of these 6 genes was detected in leaves transformed with either the empty vector (EV) or the cTP-depleted HCF152

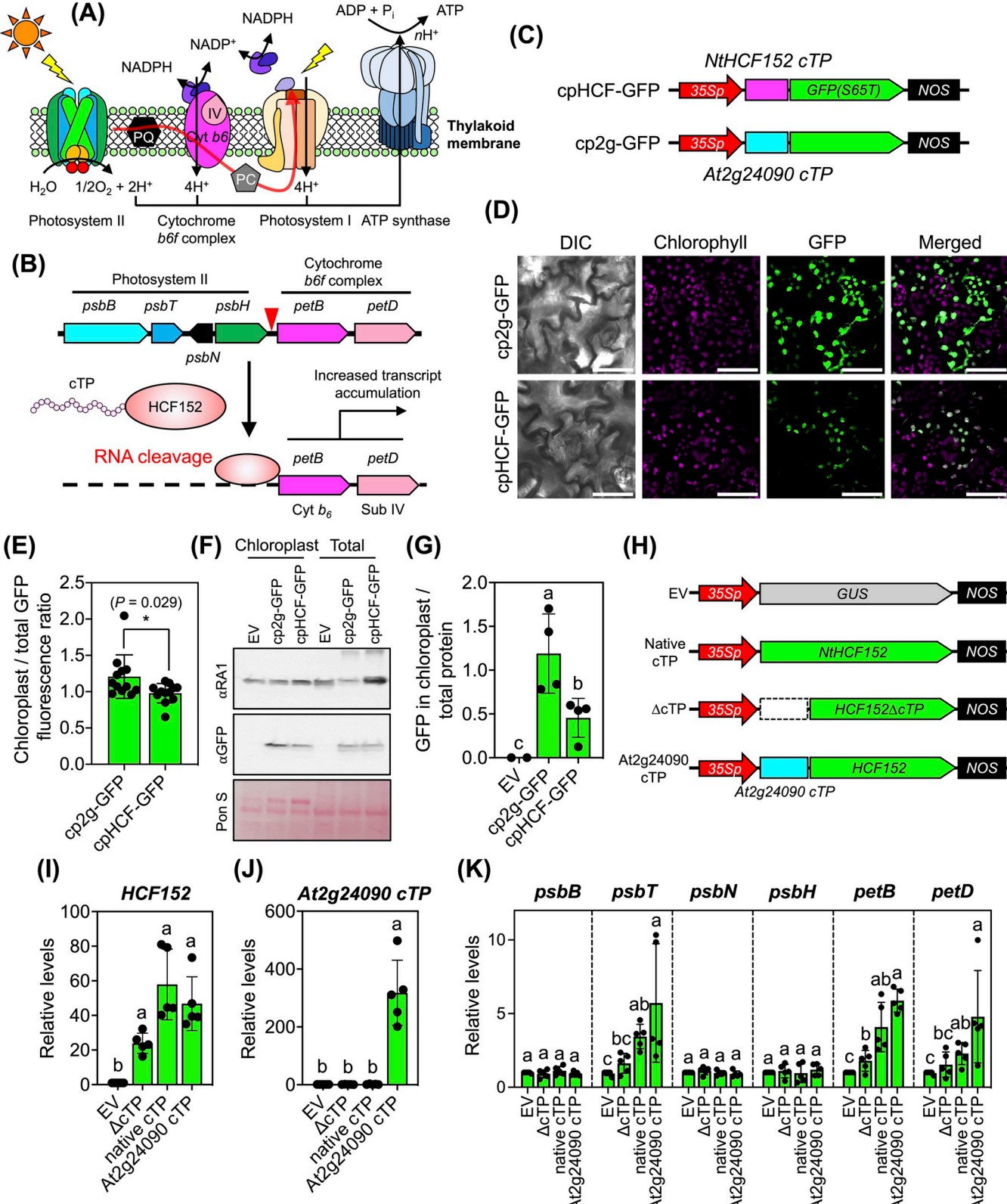

**Fig 7. cTP-mediated protein targeting increases transcript levels in chloroplast.** (**A**) Diagram of photosystems and electron transport complexes on the thylakoid membrane of the chloroplast. (**B**) RNA processing and activation of plastidial photosynthesis-related transcription by NtHCF152. (**C**) Expression cassettes of recombinant cTP-GFPs used to compare import efficiencies. (**D**) Localization of different recombinant cTP-GFPs within chloroplasts of transfected

tobacco leaf cells after agroinfiltration for 3 DAI. Scale bars = 50 μm. (**E**) Comparative fluorescence analysis of recombinant cTP-GFPs within the chloroplasts of plant cells. Error bars represent standard deviations of the mean determined from 12 CLSM images from 2 independent experiments (see fluorescence values in S17 Data). Asterisks indicate a significant difference in the mean (Student's *t* test). (**F, G**) Immunoblotting and quantitative analysis of the abundance of recombinant cTP-GFPs in total leaf proteins and chloroplast proteins. GFP-fused recombinant proteins expressed in plant cells and chloroplasts were analyzed using anti-GFP antibody. The band intensity of GFP-specific signal was quantified using Fiji ImageJ. Error bars in (**G**) represent the standard deviation of the mean from 4 independent transformation experiments ($n$ = 4, S17 Data). Different letters indicate significant differences in the mean among treatments (one-way ANOVA with Tukey's HSD test at $p$ = 0.05). (**H**) Expression constructs of cTP-engineered HCF152. EV = empty vector. ΔcTP = expression cassette for cTP-depleted NtHCF152 protein. (**I, J**) Expression of NtHCF152- and At2g24090 cTP-coding genes in tobacco leaves. (**K**) Transcript abundances of photosynthesis-related genes in tobacco leaves expressing recombinant At2g24090 cTP-HCF152 protein. Error bars represent the standard deviations of the mean from 5 transformation experiments ($n$ = 5). Different letters in (**I–K**) indicate significant differences in means (one-way ANOVA with Tukey's HSD test at $p$ = 0.05). Relative transcript expression values in **Fig 7I–7K** are available in S17 Data. CLSM, confocal laser-scanning microscopy; cTP, chloroplast-targeting peptide; DAI, days after agroinfiltration; GFP, green fluorescent protein.

vector (**Fig 7K** and S17 Data). These findings suggest that At2g24090 cTP efficiently facilitates the import of cTP-engineered NtHCF152 into chloroplasts, resulting in the enhanced stability of photosynthesis-related plastidial RNA molecules, such as the *petB* and *petD* transcripts involved in the assembly of the cytochrome b6 complex.

## Discussion

In this study, we employed both computational and experimental approaches to identify a novel set of high-efficacy cTPs from an available genomic database [29,30]. Our initial step involved predicting 88 credible polypeptides derived from the N-terminal regions of essential chloroplast proteins. We classified these sequences based on their chloroplast-targeting activities by performing in planta expression and fluorescence tracking. Our analysis unveiled the multi-organellar targeting of these cTPs into plant chloroplasts, which is consistent with previous findings [40,41]. It is important to note that variations in the expression and stability of various cTP-GFPs influenced our fluorescence-based analysis, which relied solely on the levels of fluorescent proteins within plant cells.

Exploring organellar proteins using experimental methods yields valuable insights, contributing to the refinement of prediction models. Various algorithms have been employed to train computational predictors aimed at classifying organellar proteins based on sequence data and expression/interaction features [42]. *Arabidopsis* is predicted to contain 2,090 nucleus-encoded chloroplast proteins [43]. However, our attempts to establish correlations between the scores generated by different predictors (TargetP-2.0 [32], PredSL [44], Localizer [45], and DeepLoc-1.0 [46]) and our experimental assessments of cTP targeting abilities yielded inconclusive results (S21 Fig and S18 and S19 Data). Furthermore, the high-efficiency cTPs did not exhibit a consensus cTP sequence (S22 Fig and S20 Data), except for the presence of serine residues, which is a common feature in cTPs [4,5,7,8].

Our finding that the cTPs of AtRbcS1A, At1g63970, and At2g20920 showed similar in vitro import efficiencies disproves the notion that the translocation capabilities different cTPs vary drastically. However, a meticulous inspection of the time-course expression of recombinant cTP-GFPs in agroinfiltrated plant cells highlights markedly superior abilities of the selected cTPs (At1g63970, At2g20920, and At2g24090) over the reference cTP AtRbcS1A in facilitating protein import into chloroplasts. This implies that supplementary factors or regulatory mechanisms could influence their import efficiencies in vivo, including scaffold formation with protein partners and conformational changes necessary for domain recognition during protein import into plastids. Indeed, distinguishing the translocation mechanisms of an ideal cTP that demonstrates the highest chloroplast import function may necessitate further experimental approaches. Although computational predictions provided valuable insights into organellar targeting and the optimal length of cTPs, they were less successful in estimating the precise

targeting abilities of cTPs. In practice, a combinatory approach using prediction tools like machine learning coupled with a high-throughput in planta expression platform represents a fundamental strategy for unraveling the intricate functions of transit peptides [47]. This approach could also facilitate the characterization of mitochondrion-targeting peptides (mTPs) and subplastidial-localizing peptides, such as thylakoid-targeting peptides (tTPs), from available databases.

Plastid engineering in which a high abundance of preproteins was guided to plastids by a highly efficient cTP enhanced organellar processes [18,28]. cTPs that localize to the stroma typically comprise 3 distinct regions: an uncharged amino (N)-terminal segment, a central region enriched in serine but devoid of acidic amino acids, and an amphiphilic β-stranded carboxyl (C)-terminal domain. This structural configuration helps the cTP form a scaffold with the chloroplast import machinery, but the cTP is susceptible to removal by stromal processing peptidases. We demonstrated that exchanging the precursory transit peptides with extrinsic At2g24090 cTP strongly enhanced the organellar functions of chloroplast-targeted proteins.

Although we did not detect any disruptions in protein activity when we replaced a native cTP with a novel cTP, occasional disturbances may occur due to the mistargeting of preproteins or the failure to cleave the introduced cTP, resulting in the formation of mature proteins within plastids [18,20,35,48,49]. The cleavage of At2g24090-GFP by peptidases underscores the critical importance of predictable cleavage site in this cTP to ensure proper targeting and optimize import efficiency. Switching the obstructive cleavage site of the At2g24090 transit peptide with the conserved sequence that we have currently identified holds promise for enhancing the stability of recombinant proteins against cellular peptidases. This modification could lead to heightened chloroplast import activity of the engineered cTPs, thus potentially enhancing their functional efficiency. Furthermore, our truncation analysis of At2g24090 cTP highlighted the essential role of cTP length in facilitating efficient protein translocation into chloroplasts, highlighting the importance of cTP length and cTP processing for the targeting of other plastidial proteins. Hence, the systematic creation of an extensive collection of cTPs could help support plastid engineering initiatives. Additionally, identifying motifs capable of facilitating the import mechanism of At2g24090 cTP could enhance its targeting activity by introducing mutations or performing domain swapping [13,50,51]. Moreover, conducting comparative studies involving At2g24090 cTP homologs from other plant species would expedite the construction of an optimal cTP, thereby maximizing biomolecule targeting to plastids.

The convergence of cutting-edge gene-editing tools represents a highly promising platform for advancing chloroplast engineering, with profound implications for both fundamental research and practical applications. The pursuit of modern enhancements in the economic traits of crops, facilitated by plastid-targeted base editors including ZFN [25], TALEN [23,24], and CRISPR-type deaminase technologies [22], requires the efficient delivery of functional enzymes into chloroplasts. It is increasingly evident that the gene-editing efficacies of these systems are intricately orchestrated by the import activities of the associated cTPs. In this context, a discovery of a high-efficiency cTP significantly expands the repertoire of tools available for the effective delivery of recombinant base-editing proteins to plastids. This finding supports pivotal chloroplast engineering and plastome modification mediated by cTP-engineered proteins in the premeditated crop species through non-transgenic approach using transient expression systems [52,53]. Furthermore, our work highlights the potential of conjugating exogenous chloroplast-targeting proteins with nanocarriers, enabling precise and controlled delivery into plastids [26,54,55]. Expanding the application of cTP-conjugated biomolecule delivery to plastids necessitates careful development due to potential variations in import activity among different target plant cells. Moreover, comprehensive assessment of potential side effects such as silencing of endogenous gene expression, cryptic protein formation, and

cytotoxicity of highly abundant recombinant proteins in transfected cells is essential following the introduction of cTP-engineered cargo molecules into native plant systems. At2g24090 cTP emerges as a compelling biorecognition motif for nanocarriers, offering a unique opportunity to enhance their plastid-targeting capabilities, thereby amplifying the transformative potential of this innovative fusion between gene-editing technology and advanced biomolecule carriers for chloroplast engineering.

## Materials and methods

### Computational prediction of putative cTPs

The sequences of 92 protein-encoding genes associated with 147 *A. thaliana* mutants with chloroplast-related phenotypes were obtained from the Chloroplast Functional Database (S1 Data) [29,30]. The deduced protein sequences of 88 of these genes were examined to predict the presence of putative cTP domains and the corresponding cleavage sequences at their N-termini using TargetP-2.0 (S2 Data) [32]. Multiple sequence alignment of the predicted cTPs was performed using MetaLogo [56]. Amino acid compositions of the polypeptide chains were computationally determined using the EMBOSS Pepstats workspace [57].

### Plant cultivation

*Arabidopsis* seeds were disinfected with 70% (v/v) ethanol and 20% (v/v) kitchen Haiter bleach solution (Kao, Tokyo, Japan) and germinated on 1/2 MS medium + 30 g/l sucrose + 2.5 g/l Phytagel (Sigma Aldrich, Saint Louis, Missouri, United States of America). Seedlings were grown at 22°C under a 16/8 light/dark photoperiod at 100 μmol photons m$^{-2}$·s$^{-1}$ in a growth chamber for 2 weeks. The leaves were subjected to RNA extraction.

Tobacco (*Nicotiana tabacum*) and tomato (*Solanum lycopersicum* cv. MicroTom) seeds were germinated in premixed soils (HonenAgri Co., Ltd., Niigata, Japan) at 24°C under a 12/12 hours light/dark photoperiod at 100 μmol photons m$^{-2}$·s$^{-1}$ in a growth chamber. Each two-week-old seedling was transferred and cultured individually under the same conditions. Five- to six-week-old tobacco plants with at least 3 fully expanded leaves were used for agroinfiltration. Tomato flowers were hand-pollinated to obtain a sufficient number of fully developed mature fruits. Mature green fruits were used for gene expression studies after *Agrobacterium* injection.

### Preparation of biomolecules

Total RNA was purified from plant tissues using an RNeasy Plant Mini Kit (Qiagen, Hilden, Germany). RNA samples were kept at –80°C. Complementary DNA (cDNA) molecules were synthesized from 0.5 μg of purified RNA using a ReverTra Ace qPCR RT Master Mix with gDNA Remover kit according to the manufacturer's protocol (Toyobo, Osaka, Japan). Solutions containing 0.1 μg/μl cDNA were kept at −30°C for further experiments.

Plasmid DNA was purified from liquid *Escherichia coli* cultures using a QIAprep Spin Mini Prep kit (Qiagen). Purified DNA was digested with restriction enzymes (Takara Bio, Shiga, Japan), and DNA fragments were purified through a QIAquick Gel Extraction column before ligation (Qiagen). For expression and subcellular localization studies by fluorescent tagging, DNA sequences encoding 89 cTPs were amplified from cDNA of *Arabidopsis* leaves using the cloning primers listed in S21 Data. These sequences included the predicted cleavage sites of each peptide. DNA fragments corresponding to at least the first 60 amino acids of preproteins without the predicted cTP domain were cloned and subjected to ligation. The *GFP(S65T)* gene was amplified from the DNA template using the primers listed in S21 Data. The PCR

fragments were cloned into the pTA2-cloning vector using a TArget Clone-Plus cloning kit (Toyobo). DNA fragments of the cTPs and *GFP(S65T)* in the cloning vector were digested with the restriction enzymes *Xba*I/*Bam*HI and *Bam*HI/*Sac*I, respectively. Ligation fragments were then ligated to the *Xba*I/*Sac*I-linearized pBI121 vector fragments with a Ligation High Ver2.0 DNA ligation kit (Toyobo) to create pBI121-*cTP*::*GFP* expression vectors. The ligation products were transformed into *E. coli* DH5α competent cells.

The coding regions of *Arabidopsis AtGGPPS2* and *AtPSY1* were amplified using cDNA from *Arabidopsis* leaves using the primers listed in S22 Data. The coding region of tobacco NtHCF152 was cloned from tobacco cDNA with the primers listed in S22 Data. These DNA fragments were subcloned into *Xba*I/*Sac*I-linearized pBI121 DNA fragments. DNA fragments of the cTP domains and cTP-depleted coding sequences of each protein were cloned from the resulting expression vectors. The coding sequences of cTPs were subcloned into the pBI121-*At2g24090 cTP*::*GFP* expression vector at the *Xba*I/*Bam*HI sites. The cTP-depleted coding regions of these 3 proteins were replaced with *GFP(S65T)* in the pBI121-*At2g24090 cTP*::*GFP* expression cassette at the *Bam*HI/*Sac*I sites. The resulting expression vectors were transformed and maintained in *E. coli* DH5α cells.

### *Agrobacterium*-mediated gene transfer

Plant expression vectors were transformed into *Agrobacterium tumefaciens* strain LBA4404 using electroporation. Briefly, competent *Agrobacterium* cells were mixed with 0.1 µg of plasmid DNA. Electroporation was performed with a MicroPulser Electroporator following the manufacturer's protocol (Bio-Rad Laboratories, Tokyo, Japan).

A single colony of *Agrobacterium* harboring a plant expression vector was cultured in 10 ml of YEB medium + 50 mg/l kanamycin + 20 µm acetosyringone at 30˚C and 180 rpm overnight. *Agrobacterium* cells were collected by centrifugation and resuspended in 10 mM MES buffer, pH 5.7 + 10 mM MgCl$_2$ + 200 µm acetosyringone (i.e., agroinfiltration buffer). The optical density (OD) of the *Agrobacterium* suspensions was adjusted to OD$_{600nm}$ = 0.2 with agroinfiltration buffer, and the cultures were incubated in the dark at ambient temperature for 2 h. Fully expanded tobacco leaves were infiltrated with *Agrobacterium* solution using a 1 ml needle-less syringe. Intact mature green tomato (cv. MicroTom) fruits were injected with *Agrobacterium* solution using a 1 ml syringe attached to a 27-gauge needle. Transformed plants were cultured under standard conditions for further analysis.

### Fluorescence microscopy and image analysis

Recombinant cTP-GFP expressed in plant cells was observed under a confocal laser-scanning microscope (CLSM; Zeiss LSM700, Carl Zeiss, Oberkochen, Germany). The excitation/emission (ex/em) wavelengths for the detection of cTP-GFP fluorescent proteins were 488/510–525 nm. Chlorophyll autofluorescence in plant cells was detected at ex/em of 488/640–700 nm. Fluorescence imaging was conducted on the abaxial sides of tobacco leaves at 3 days post-infiltration. For screening and subcellular localization studies, fluorescence images were taken under a Plan-Apochromat 20×/0.8 M27 objective lens with a digital zoom of 2. The laser power from the diode source was preset at 10, and the digital gains of GFP and chlorophyll fluorescence were 550 and 600, respectively. The size of the 16-bit image was 1,024 × 1,024 pixels. For comparative analysis of fluorescence images, the digital gain of GFP was reduced to 500 to increase the detection sensitivity of GFP signals in plant cells. At least 5 regions of interest (ROIs) were taken per leaf per experiment.

GFP and chlorophyll fluorescence was quantified using Fiji ImageJ [58]. A fluorescence image was converted to 8-bit grayscale format. The background intensity in the image was

corrected by setting the lower and upper threshold levels to 30 and 255, respectively, before quantifying the total GFP signal in an ROI [59]. Fluorescence from GFP and chlorophyll in chloroplasts was measured using the Particles analysis plug-in, with the range of particle sizes in an ROI set to 5 to 40 $\mu m^2$. The fluorescence intensities of GFP and chlorophyll in a given number of chloroplasts in an ROI were equal to the mean fluorescence multiplied by the integrated density of the grayscale value (Mean × IntDen). Total GFP fluorescence in plant cells and GFP signals in chloroplasts were normalized by the chlorophyll autofluorescence in chloroplasts per ROI.

## Fluorescence measurement

For the measurement of recombinant fluorescent protein content, total leaf proteins were extracted from agroinfiltrated tobacco leaves collected at different hours post infiltration using 100 mM Tris-HCl (pH 7.4) + 150 mM NaCl + 0.1% (v/v) Triton X-100 at 20,380 × g for 15 min. One-hundred microliters of the sample were applied to a well of a FluoTrac 96-well microplate (Greiner Bio-One, Tokyo, Japan). Various amounts of standard GFP purified from *E. coli* cells were utilized to generate a linear regression equation for calculating the fluorescent protein content in the samples. Fluorescence in total proteins was determined using the fluorescence mode in a Spark 10M microplate reader (Tecan, Männedorf, Switzerland). The concentration of total leaf proteins was measured using Bradford reagent with bovine serum albumin (BSA) as the standard (Bio-Rad).

## Immunoblot analysis

Total leaf proteins were extracted from agroinfiltrated tobacco leaves with 6 M urea cracking solution (6 M urea + 100 mM Tris-HCl, pH 7.0 + 20% (v/v) glycerol + 10% (w/v) SDS + 5% (v/v) β-mercaptoethanol + 1 tablet of cOmplete, EDTA-free Protease Inhibitor Cocktail (Roche Diagnostics GmBH, Manheim, Germany)). Chloroplasts were isolated from tobacco leaves in a 40% (v/v) Percoll gradient. Briefly, leaf segments (0.5 × 0.5 cm) were ground in 5 ml of ice-cold chloroplast isolation buffer (0.33 M sorbitol, 0.1 M Tris-HCl, pH 7.5, 5 mM $MgCl_2$, 10 mM NaCl, 2 mM EDTA, pH 8.0, and 0.1% (w/v) BSA) with a prechilled mortar and pestle on ice. The chloroplast solutions were filtered through a layer of Miracloth with a pore size of 22 to 25 μm (Merck MGaK, Darmstadt, Germany). The filtrates were overlaid on 5 ml of 40% (v/v) Percoll (Sigma Aldrich) diluted in chloroplast isolation solution. The gradient solutions were centrifuged at 1,700 × g at 4°C for 8 min. The chloroplast pellet at the bottom of the centrifuge tube was recovered and washed once with chloroplast isolation solution by centrifugation. The chloroplast proteins were extracted from isolated chloroplasts using 6 M urea cracking solution.

Ten micrograms of total leaf proteins and isolated chloroplast proteins were resolved in a Mini PROTEAN TGX Precast gel and blotted onto a 0.45 μm PVDF membrane using a Tran-Blot SD Semi-Dry Transfer Cell (Bio-Rad). GFP bands on the membrane were probed with rabbit anti-GFP polyclonal antibodies (NB600-308; Novus Biologicals, Littleton, Colorado, USA) as the primary antibody at a dilution of 1:5,000. Horseradish peroxidase (HRP)-conjugated goat anti-rabbit IgG polyclonal antibody (ab6721; Abcam, Tokyo, Japan) at a dilution of 1:20,000 was used as the secondary antibody. To detect FLAG-tagged recombinant proteins, the immunoblot membranes were incubated with primary antibody solution containing 1:10,000 mouse anti-FLAG antibody (OctA-probe (H-5); Santa Cruz Biotechnology, Texas, USA). The secondary antibody used for detection was an HRP-conjugated goat anti-mouse IgG antibody (ab6789, Abcam) at a dilution of 1:50,000. After probing with secondary antibody, 1 ml of SuperSignal West Pico PLUS chemiluminescence substrate mixture (Life

Technologies, Carlsbad, California, USA) was applied to the membrane prior to detecting the HRP signal using an LAS3000 imaging system (Fujifilm, Tokyo, Japan). After GFP detection, the membrane was stripped using membrane stripping solution (62.5 mM Tris-HCl, pH 6.8 + 2% (w/v) SDS + 100 mM β-mercaptoethanol) at 50˚C for 1 h and incubated with an antibody solution containing 1:5,000 rabbit anti-Rubisco activase 1 (RA1) polyclonal antibody (AS10700; Agrisera, Vännäs, Sweden). The secondary antibody solution was produced and anti-RA1 antibody on the membrane detected as described for the anti-GFP antibody. The band intensity of the target protein on the membrane was quantified using Fiji ImageJ [58].

## Preparation of recombinant cTP-GFP for in vitro import analysis

Recombinant cTP-GFP DNA fragments amplified with PCR primers listed in S23 Data were subcloned into the pET-15b vector at the *Nde*I/*Xho*I sites. *E. coli* strain BL21(DE3) was transformed with these expression vectors for protein expression and purification. Bacterial cells were cultured in LB medium at 37˚C until reaching $OD_{600nm} = 0.8$. Recombinant cTP-GFP expression in *E. coli* was induced with 0.5 mM isopropyl β-D-1-thiogalactopyronoside (IPTG) at 16˚C for 12 to 24 h. After induction, the bacterial cells were harvested by centrifugation at $1,450 \times g$ for 15 min, snap-frozen in liquid nitrogen, and stored at –80˚C after. Soluble protein fractions from *E. coli* cells were obtained using cell denaturation buffer (8 M urea, 10 mM Tris-HCl (pH 7.4), 150 mM NaCl, 0.1% Triton X-100) and then centrifuged at $20,380 \times g$ for 30 min. The soluble protein fractions were incubated with Ni-NTA agarose resin (Qiagen) in cell denaturation buffer at room temperature for 2 h. The Ni-NTA agarose resin was washed 10 times with 1 ml washing solution (10 mM Tris-HCl (pH 7.4), 150 mM NaCl, 20 mM imidazole). Purified 6xHis-cTP-GFP was eluted from the Ni-NTA column using elution buffer (10 mM Tris-HCl (pH 7.4), 150 mM NaCl, 200 mM imidazole) and desalted using an Amicon Ultra Centrifugal Filter Unit (Merck). N-terminal His-tag polypeptides of recombinant cTP-GFP were removed with a Thrombin Cleavage kit (Abcam), and the concentration of purified protein was determined using Bradford reagent with BSA as the standard (Bio-Rad).

In vitro synthesis of recombinant cTP-GFP protein was conducted using the $T_NT$ Coupled Wheat Germ Extract System following the manufacturer's protocol (Promega). Five micrograms of *Eco*RI-linearized pET15b-cTP-GFP expression vectors served as the DNA templates for in vitro transcription and cell-free protein synthesis. Analysis of 1 μg of *E. coli*-derived purified recombinant proteins or 2 μl of cell-free synthesis reactions was carried out on a 14% SDS-PAGE gel at 100 volts for 1 h, followed by staining with Coomassie brilliant blue (CBB) solution and immunoblotting.

## In vitro chloroplast import assay

The differential efficiencies of selected recombinant cTP-GFPs were examined using in vitro chloroplast import assays [31]. In brief, 100 nM of *E. coli*-derived recombinant cTP-GFP was incubated with isolated tobacco chloroplasts in import (HS) solution containing 20 mM gluconic acid, 10 mM NaHCO$_3$, 0.2% (w/v) BSA, and 5 mM MgATP. The import reactions were conducted under white light (100 μmol m$^{-2}$ s$^{-1}$) at 25˚C. For time-course analysis, 50 μl of the import reaction was collected at various time points, and an equal volume of 6 M urea cracking buffer was added to the reaction to lyse the chloroplasts on ice for 30 min. Subsequently, soluble proteins in the import reaction were isolated by centrifugation at $20,380 \times g$, 4˚C, for 20 min, and 2 μg of the protein sample was used for immunoblot analysis using anti-GFP antibody as described above.

## qRT-PCR analysis

Fifty nanograms of cDNA sample was used as a template for quantitative reverse-transcription PCR (qRT-PCR) using PowerUp SYBR Green Master Mix (Thermo Fischer Scientific Baltics UAB, Vilnius, Lithuania) with the gene-specific primers listed in S22 Data. The constitutively expressed housekeeping gene tobacco *ACTIN4* (*NtACT4*) was used as an expression control for both tobacco and tomato (S22 Data). The comparative $C_T$ ($2^{-\Delta\Delta C_T}$) method was used to compare differential gene expression in plant tissues [60].

## Analysis of pigment contents

Chlorophyll and carotenoid contents in plant tissues were analyzed by spectrophotometry [61]. Briefly, tobacco leaves and tomato fruits were ground in liquid nitrogen, and pigments were extracted from the ground tissues with 80% (v/v) cold acetone in water, followed by centrifugation at $20,380 \times g$ at 4°C for 15 min. The absorbance of the acetone fraction was analyzed at wavelengths of 663 nm for chlorophyll a, 646 nm for chlorophyll b, and 470 nm for carotenoids. The levels of each pigment in the acetone fractions were determined using previously reported equations for 80% aqueous acetone as the solvent [61].

## Statistical analysis

The distribution of experimental data from at least 2 independent experiments was presented as box plots created using R Studio (R Studio, Boston, Massachusetts, USA). The black bars in the box plots represent the median of the distributed values. Each dot represents a data point. Unless otherwise mentioned, statistically significant differences among treatments were analyzed by ANOVA using Jamovi version 1.6 (The Jamovi Project, Sydney, Australia) or Graph-Pad Prism 10 (GraphPad Software, Boston, Massachusetts, USA). The chloroplast-targeting efficiencies of cTP-GFP based on chloroplast GFP/chlorophyll and total GFP/chlorophyll fluorescence were examined by *K*-means clustering with Hartigan–Wong's algorithm in the Snowcluster plug-in of Jamovi 1.6.

## Supporting information

**S1 Fig. Computational prediction of signal peptides and their cleavage sites in 89 polypeptide sequences. (A)** Distribution of mTP scores in 89 polypeptides predicted by TargetP-2.0. The mTP score of each transit peptide (as presented in S3 Data) is shown as a box plot. Magenta dots represent the mTP scores of each peptide predicted to target the chloroplast (cTP), thylakoid lumen (tTP), and other organelles in plant cells (other). Significant differences are indicated by different letters in the boxplot (one-way ANOVA with Tukey's HSD test at $p = 0.05$). **(B)** Amino acid distribution in 13 amino acids, which encompassed the predicted cleavage sites of each transit peptides (cTPs). **(C)** Alignment of the predicted cleavage sites of 8 tTPs. Multiple sequence alignments were performed using WebLogo. Five amino acids positioned within the magenta box at −3 to 1 represent the predicted cleavage site of the cTPs or tTPs with an extension of 4 amino acids applied to both its N- (−7 to −4) and C-termini (2 to 5). Amino acid at position 0 is the cleaved amino acid, which is the last amino acid attached to each transit peptide after cleavage.
(JPG)

**S2 Fig. Expression and subcellular localization of different cTP-GFP(S65T) proteins in tobacco leaf cells.** Subcellular localizations of different cTP-GFP(S65T) proteins in tobacco leaf cells at 4 days post agroinfiltration were observed under a confocal laser-scanning microscope (LSM 700). Scale bars = 20 μm. Immunoblot analysis of total leaf proteins (T) and

isolated chloroplast proteins (C) showing the abundances of cTP-GFP(S65T) proteins in the cytosol and chloroplasts after translocation. αGFP = immunoblotted membranes probed by anti-GFP antibody. PS = Ponceau S-stained membrane to show the equal loading of protein samples.
(JPG)

**S3 Fig. Expression and subcellular localization of different cTP-GFP(S65T) proteins in tobacco leaf cells.** Subcellular localizations of different cTP-GFP(S65T) proteins in tobacco leaf cells at 4 days post agroinfiltration were observed under a confocal laser-scanning microscope (LSM 700). Scale bars = 20 μm. Immunoblot analysis of total leaf proteins (T) and isolated chloroplast proteins (C) showing the abundances of cTP-GFP(S65T) proteins in the cytosol and chloroplasts after translocation. αGFP = immunoblotted membranes probed by anti-GFP antibody. PS = Ponceau S-stained membrane to show the equal loading of protein samples.
(JPG)

**S4 Fig. Expression and subcellular localization of different cTP-GFP(S65T) proteins in tobacco leaf cells.** Subcellular localizations of different cTP-GFP(S65T) proteins in tobacco leaf cells at 4 days post agroinfiltration were observed under a confocal laser-scanning microscope (LSM 700). Scale bars = 20 μm. Immunoblot analysis of total leaf proteins (T) and isolated chloroplast proteins (C) showing the abundances of cTP-GFP(S65T) proteins in the cytosol and chloroplasts after translocation. αGFP = immunoblotted membranes probed by anti-GFP antibody. PS = Ponceau S-stained membrane to show the equal loading of protein samples.
(JPG)

**S5 Fig. Expression and subcellular localization of different cTP-GFP(S65T) proteins in tobacco leaf cells.** Subcellular localizations of different cTP-GFP(S65T) proteins in tobacco leaf cells at 4 days post agroinfiltration were observed under a confocal laser-scanning microscope (LSM 700). Scale bars = 20 μm. Immunoblot analysis of total leaf proteins (T) and isolated chloroplast proteins (C) showing the abundances of cTP-GFP(S65T) proteins in the cytosol and chloroplasts after translocation. αGFP = immunoblotted membranes probed by anti-GFP antibody. PS = Ponceau S-stained membrane to show the equal loading of protein samples.
(JPG)

**S6 Fig. Expression and subcellular localization of different cTP-GFP(S65T) proteins in tobacco leaf cells.** Subcellular localizations of different cTP-GFP(S65T) proteins in tobacco leaf cells at 4 days post agroinfiltration were observed under a confocal laser-scanning microscope (LSM 700). Scale bars = 20 μm. Immunoblot analysis of total leaf proteins (T) and isolated chloroplast proteins (C) showing the abundances of cTP-GFP(S65T) proteins in the cytosol and chloroplasts after translocation. αGFP = immunoblotted membranes probed by anti-GFP antibody. PS = Ponceau S-stained membrane to show the equal loading of protein samples.
(JPG)

**S7 Fig. Expression and subcellular localization of different cTP-GFP(S65T) proteins in tobacco leaf cells.** Subcellular localizations of different cTP-GFP(S65T) proteins in tobacco leaf cells at 4 days post agroinfiltration were observed under a confocal laser-scanning microscope (LSM 700). Scale bars = 20 μm. Immunoblot analysis of total leaf proteins (T) and isolated chloroplast proteins (C) showing the abundances of cTP-GFP(S65T) proteins in the

cytosol and chloroplasts after translocation. αGFP = immunoblotted membranes probed by anti-GFP antibody. PS = Ponceau S-stained membrane to show the equal loading of protein samples.
(JPG)

**S8 Fig. Expression and subcellular localization of different cTP-GFP(S65T) proteins in tobacco leaf cells.** Subcellular localizations of different cTP-GFP(S65T) proteins in tobacco leaf cells at 4 days post agroinfiltration were observed under a confocal laser-scanning microscope (LSM 700). Scale bars = 20 μm. Immunoblot analysis of total leaf proteins (T) and isolated chloroplast proteins (C) showing the abundances of cTP-GFP(S65T) proteins in the cytosol and chloroplasts after translocation. αGFP = immunoblotted membranes probed by anti-GFP antibody. PS = Ponceau S-stained membrane to show the equal loading of protein samples.
(JPG)

**S9 Fig. Quantitative GFP fluorescence measurement of recombinant cTP-GFP in plant cells.** (**A**) Distribution of normalized GFP fluorescence in plant cells transiently expressing recombinant fluorescent protein. (**B**) Normalized GFP fluorescence in chloroplasts of plant cells transformed with 89 different *cTP-GFP* expression cassettes. Ten independent CLSM images ($n$ = 10) were taken from 2 tobacco leaves from 2 independent experiments. GFP fluorescence in plant cells and chloroplasts and chlorophyll autofluorescence were quantified by Fiji ImageJ. Normalized fluorescence values in each treatment are presented as box plots. Magenta circles represent the distribution of data in the box plot. Black bars are medians. Fluorescence values in S9A and S9B Fig can be also found in S6 Data.
(JPG)

**S10 Fig. Distribution of amino acids in the 13 residues surrounding the predicted cleavage sites of each transit peptide (cTPs).** The alignment encompasses the cleavage sites of cTPs demonstrating chloroplast-specific targeting of GFP (Chl), dual-targeting cTPs that transport GFP to both chloroplasts and mitochondria (Chl/Mt) or to the cytosol (Chl/Cy), and the predicted cTPs that mislocalize GFP to the cytosol instead of the chloroplast (Cy). Multiple sequence alignments were conducted using WebLogo. The 5 amino acids within the magenta box at positions −3 to 1 represent the predicted cleavage site of the cTPs or tTPs, along with an additional 4 amino acids on both the N-terminus (−7 to −4) and C-terminus (2 to 5). The amino acid at position 0 corresponds to the cleaved residue, which constitutes the final amino acid attached to each transit peptide after cleavage. The number in the brackets denotes the count of cleavage sequences of the predicted cTPs in each group.
(JPG)

**S11 Fig. Classification of cTP-GFP preproteins.** Recombinant cTP-GFP was transiently expressed in tobacco leaf cells by Agroinfiltration. CLSM imaging and fluorescence measurement were performed at 3 DAI. (**A**) Correlation between total GFP fluorescence in CLSM images of leaf cells and GFP signals inside the chloroplasts. GFP fluorescence in plant cells and chloroplasts was normalized by the respective chlorophyll autofluorescence in the image. (**B**) Eighty-nine cTP-GFP proteins were classified into 4 different clusters based on the correlation between total GFP in plant cells and GFP fluorescence in chloroplasts. Fluorescence values and classification of different cTP-GFPs are presented in S7 Data.
(JPG)

**S12 Fig. Immunoblotting of cTP-GFP preproteins in agroinfiltrated tobacco leaves.** Differential accumulation of total leaf proteins and isolated chloroplast proteins in agroinfiltrated

leaves was analyzed by SDS-PAGE and immunoblot analysis using anti-GFP (αGFP) and anti-RA1 (αRA1) antibodies. Three independent experiments were conducted. Band intensities of the target proteins of the respective sizes were quantified by Fiji ImageJ. Quantitative immunoblot results and representative images are shown in **Fig 2**.
(JPG)

**S13 Fig. In vitro import analysis of recombinant cTP-GFP.** Purified recombinant cTP-GFPs (obtained from the *E. coli* expression system) were incubated with isolated tobacco chloroplasts. Import assay reactions were collected at various time points following incubation, and total proteins in the import reactions were separated using 6 M urea cracking solution. Subsequently, 2 μg protein samples were subjected to immunoblotting with anti-GFP antibody. After immunoblotting, the membranes were stained with CBB. The band intensities of the GFP-specific signal on the membrane were quantified using Fiji ImageJ. To ensure statistical validity, 3 independent import experiments were conducted.
(JPG)

**S14 Fig. Ratio of GFP fluorescence in chloroplasts to total GFP in tobacco leaf cells after agroinfiltration.** GFP fluorescence in plant cells and within chloroplasts was quantified using Fiji ImageJ in confocal laser scanning microscopy (CLSM) images of transfected tobacco leaf cells expressing different recombinant cTP-GFP constructs. The box plot illustrates the distribution of fluorescence ratios across 16 CLSM images ($n = 16$, S11 Data) collected from 3 biologically independent leaves at various time points post agroinfiltration. The central bars indicate the median fluorescence ratios at each time point, while the upper and lower bars depict the maximum and minimum values, respectively. Asterisks denote different levels of statistical significance in the mean fluorescence ratios at each time point compared to AtRbcS1A-GFP, used as a control (*; $P \leq 0.05$, **; $P \leq 0.01$, ***; $P \leq 0.001$, and ****; $P < 0.0001$). "n. s." indicates no significant difference in mean fluorescence ratio compared to the control.
(JPG)

**S15 Fig. In vivo transport of recombinant cTP-GFP to chloroplasts.** The transport of recombinant cTP-GFP to chloroplasts in transfected tobacco leaf cells was analyzed at 24, 36, 48, 60, 72, and 96 h after agroinfiltration using immunoblotting. The presence of various cTP-GFP precursors (Pre) and their chloroplast-localized cleaved products (Mat) in total leaf proteins was detected using an anti-GFP polyclonal antibody. The intensity of the GFP band in each sample was then compared to a linear regression equation generated using different amounts of standard GFP. nnL indicates the major Rubisco large subunit protein band on the immunoblot membrane stained with CBB.
(JPG)

**S16 Fig. In vitro import analysis of truncated At2g24090 cTPs.** (**A**) SDS-PAGE of various truncated At2g24090 cTP-GFP fusion proteins. Recombinant cTP-GFPs were produced using the *E. coli* expression system, and 500 ng processed recombinant protein samples were analyzed using a 14% SDS-PAGE gel. The gel was stained with CBB to visualize the protein bands. (**B**) In vitro import assays of various truncated At2g24090 cTP-GFP fusion proteins into isolated tobacco chloroplasts. Each 100 μm recombinant protein sample was incubated with isolated chloroplasts for 3 h. Following incubation, the import reaction was halted by adding EDTA to a final concentration of 50 mM. Total proteins in the import reaction were separated using 6 M urea cracking solution, and 2 μg of protein was subjected to immunoblotting using an anti-GFP antibody. After immunoblotting, the membranes were stained with CBB. Three independent analyses were conducted to ensure statistical validity.
(JPG)

**S17 Fig. Truncated At2g24090 cTP-GFPs display different subcellular localizations in plant cells.** (**A**) Subcellular localizations of different truncated At2g24090 cTP-GFPs in tobacco leaf cells at 3 DAI. Scale bars = 50 μm. (**B**) Immunoblot analysis of total leaf proteins and isolated chloroplast proteins from tobacco leaves using anti-GFP antibody.
(JPG)

**S18 Fig. Import efficiencies of AtGGPPS2 cTP and AtPSY1 cTP to transport GFP into chloroplasts.** (**A**) Expression cassettes of recombinant cTP-GFPs in plant cells. The T-DNA vectors harboring these gene expression cassettes were introduced into tobacco leaf cells via Agroinfiltration. (**B**) Subcellular localization of different cTP-GFPs in tobacco leaf cells at 3 DAI. Scale bars = 20 μm. (**C**) Comparative analysis of cTP-GFP fluorescence in CLSM images. Distribution of GFP/chlorophyll fluorescence values in 16 ROIs from 4 different leaves is presented as a box plot (see values in S13 Data). Magenta dots represent each data point. Black bars are medians. Different letters indicate significant differences in the means among treatments (one-way ANOVA with Tukey's HSD test at $p = 0.00001$). (**D**) Immunoblot analysis of total leaf proteins and isolated chloroplast proteins from tobacco leaves infiltrated with different cTP-GFP expression vectors. RbcL bands indicate equal loading of proteins on the membrane after Ponceau S staining (PS) before immunoblotting with anti-RA1 and anti-GFP antibodies. EV = proteins from plant leaves infiltrated with *Agrobacterium* harboring pBI121-empty vector.
(JPG)

**S19 Fig. Comparative analysis of the import efficiencies of different cTPs to transport recombinant enzymes into chloroplasts.** (**A**) Expression constructs used to overexpress recombinant cTP-modified AtGGPPS2 enzymes in plants. The coding sequence of native AtGGPPS2, including its cTP portion, was fused with a FLAG tag at the C-terminus for immunoblot analysis after expression in tobacco leaf cells via Agroinfiltration. Subsequently, the cTP portion was replaced by At2g24090 cTP to potentially enhance the import efficiency of the recombinant At2g24090 cTP-GGPPS2-FLAG enzyme into the chloroplasts. (**B**) Immunoblot analysis of total leaf proteins (T) and isolated chloroplast proteins (C) in agroinfiltrated tobacco leaves. Proteins were collected from 3 biologically independent samples and stored at −80˚C. These proteins were subjected to immunoblotting using anti-FLAG antibody. The intensity of protein bands corresponding to the expected recombinant enzymes was determined using Fiji ImageJ. The membrane was stained with Ponceau S solution to ensure the equal loading of protein samples onto the membrane. (**C**) Accumulation of recombinant cTP-engineered GGPPS2-FLAG enzymes in chloroplasts analyzed by immunoblotting. An asterisk indicates a significant difference in the translocation of recombinant enzymes into chloroplasts between AtGGPPS2 cTP and At2g24090 cTP (Student's *t* test: $p < 0.05$, $n = 3$, see values in S14 Data).
(JPG)

**S20 Fig. Changes in leaf color after agroinfiltration with plant expression vectors.** (**A**) Changes in the color of agroinfiltrated tobacco leaves after incubation. Fully expanded tobacco leaves were infiltrated with *Agrobacterium* containing different plant expression vectors. (**B**) Pigment concentrations in agroinfiltrated leaves at 5 DAI. Leaf pigments in the aqueous acetone fractions were analyzed by spectrophotometry. (**C**) Transcript levels of transgenes in agroinfiltrated tobacco leaves at 5 DAI. EV = empty vector, ΔcTP = cTP-depleted recombinant proteins, WT = native proteins, 2g = At2g24090 cTP-engineered proteins. Coexpression of native- or At2g24090 cTP-engineered enzymes is indicated by WT-GGPPS2/PSY1 or 2g-

GGPPS2/PSY1, respectively. Error bars = standard deviations ($n$ = 3). Significant differences in the means compared to EV are indicated by asterisks (Student's $t$ test at $p < 0.05$). Numerical values for S20B and S20C Figs can be found in S15 Data.
(JPG)

**S21 Fig. Correlations between predicted cTP scores and chloroplast import efficiencies.** Correlations between cTP scores predicted by TargetP-2.0 (**A**), PredSL (**B**), Localizer (**C**), and DeepLoc-1.0 (**D**) and chloroplast import efficiencies of 89 transit peptide-GFP proteins (analyzed based on fluorescence measurements in CLSM images of plant cells overexpressing different cTP-GFPs) determined by linear regression. Distributions of chloroplast GFP/chlorophyll fluorescence and cTP scores are shown as density plots in (**A**–**D**). Shaded areas in the scatter plots represent standard error. The cTP scores of each preprotein from different predictions are shown in S18 and S19 Data. (**E**) Pearson's correlations between cTP scores from each computational predictor and GFP/chlorophyll fluorescence. (**F**) Statistical significances of Pearson's correlations in (**E**).
(JPG)

**S22 Fig. Multiple sequence alignment of cTPs and amino acid compositions.** (**A**) MetaLogo alignment of all 89 amino acid sequences. (**B**) Multiple sequence alignment of 10 selected cTPs for comparative analysis. (**C**) Distribution of natural amino acids in all 89 polypeptides shown as a box plot. Black bar indicates the median of the distributed values. Dots represent data points of amino acid composition in each cTP as in S20 Data.
(JPG)

**S1 Data. Chloroplast-related *Arabidopsis* mutants in the Chloroplast Functional Database.**
(XLSX)

**S2 Data. Computational prediction of 89 selected nucleus-encoded proteins related to *Arabidopsis* chloroplast mutants by TargetP-2.0.**
(XLSX)

**S3 Data. Predicted cTP scores, tTP scores, mTP scores, and amino acid length of 89 transit peptides.**
(XLSX)

**S4 Data. Relative fluorescence measurements and subcellular localizations of various cTP-GFPs in plant cells.**
(XLSX)

**S5 Data. Quantitative measurements of GFP fluorescence in different compartments in transfected plant cells.**
(XLSX)

**S6 Data. Total GFP- and chloroplast GFP fluorescence values in plant cells transformed with 89 cTP-GFP expression vectors.**
(XLSX)

**S7 Data. Average fluorescence values and classification of 89 cTP-GFPs regarding the correlation of GFP fluorescence within the chloroplasts and in plant cells.**
(XLSX)

**S8 Data. Top 10 cTPs selected for subsequent comparative studies.**
(XLSX)

**S9 Data. Quantitative measurements of CLSM imaging and immunoblotting of selected cTP-GFPs in tobacco leaf cells.**
(XLSX)

**S10 Data. In vitro import efficiency and import rate of recombinant cTP-GFPs to isolated tobacco chloroplasts.**
(XLSX)

**S11 Data. Transcript expression and GFP accumulation in agroinfiltrated tobacco leaf, time-course chloroplast targeting, import efficiency, and maturation of different recombinant cTP-GFPs in plant cells.**
(XLSX)

**S12 Data. In vitro- and in vivo import activities of truncated At2g24090-GFPs to tobacco chloroplasts.**
(XLSX)

**S13 Data. GFP fluorescence values in chloroplasts and in tobacco leaf cells transformed with different cTP-GFPs expression vectors.**
(XLSX)

**S14 Data. Accumulation of cTP-engineered GGPPS2-FLAG proteins in transformed tobacco leaf cells.**
(XLSX)

**S15 Data. Accumulation of plant pigments and transcript expression in transformed tobacco leaves expressing cTP-engineered enzymes.**
(XLSX)

**S16 Data. Relative gene expression and pigments contents in agroinfiltrated tomato fruits.**
(XLSX)

**S17 Data. Expression of recombinant At2g24090-HCF152 proteins and expression of plastidial genes involved in photosynthesis in transformed tobacco leaves.**
(XLSX)

**S18 Data. Computational predictions of targeting peptides in 89 preproteins using 4 computational prediction tools.**
(XLSX)

**S19 Data. cTP scores predicted by TargetP-2.0, PredSL, Localizer, and DeepLoc-1.0.**
(XLSX)

**S20 Data. Distribution of natural amino acid in 89 transit peptides.**
(XLSX)

**S21 Data. Primers for cloning cTP-coding sequences from Arabidopsis cDNA.**
(XLSX)

**S22 Data. Primers for cloning the coding sequences of chloroplast-targeting proteins and qRT–PCR analysis of transcript levels.**
(XLSX)

**S23 Data. Primers for constructing expression vectors for recombinant protein expression in *E. coli*.**
(XLSX)

**S1 Raw images. Original blot images for S2–S8 Figs.**
(PDF)

## Author Contributions

**Conceptualization:** Chonprakun Thagun, Yutaka Kodama, Keiji Numata.

**Funding acquisition:** Keiji Numata.

**Investigation:** Chonprakun Thagun, Masaki Odahara.

**Methodology:** Chonprakun Thagun, Yutaka Kodama, Keiji Numata.

**Supervision:** Yutaka Kodama, Keiji Numata.

**Validation:** Chonprakun Thagun, Yutaka Kodama, Keiji Numata.

**Writing – original draft:** Chonprakun Thagun, Yutaka Kodama, Keiji Numata.

**Writing – review & editing:** Chonprakun Thagun, Masaki Odahara, Yutaka Kodama, Keiji Numata.

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
