## [Editor Report · Decision Letter 0]

8 Jan 2024

Dear Dr Numata, 

Thank you for submitting your manuscript entitled "Identification of a highly efficient chloroplast-targeting peptide for plastid engineering" for consideration as a Research Article by PLOS Biology.

Your manuscript has now been evaluated by the PLOS Biology editorial staff as well as by an academic editor with relevant expertise and I am writing to let you know that we would like to send your submission out for external peer review. However, we would like to consider your manuscript as a Methods and Resources type of article, thus when you submit the metadata (see below), please select the new type of article from the dropdown menu.

Before we can send your manuscript to reviewers, we need you to complete your submission by providing the metadata that is required for full assessment. To this end, please login to Editorial Manager where you will find the paper in the 'Submissions Needing Revisions' folder on your homepage. Please click 'Revise Submission' from the Action Links and complete all additional questions in the submission questionnaire.

Once your full submission is complete, your paper will undergo a series of checks in preparation for peer review. After your manuscript has passed the checks it will be sent out for review. To provide the metadata for your submission, please Login to Editorial Manager (https://www.editorialmanager.com/pbiology) within two working days, i.e. by Jan 10 2024 11:59PM.

Kind regards,

Ines

--

Ines Alvarez-Garcia, PhD

Senior Editor

PLOS Biology

---

## [Decision Letter · Decision Letter 1]

13 Mar 2024

Dear Dr Numata,

Thank you for your patience while your manuscript entitled "Identification of a highly efficient chloroplast-targeting peptide for plastid engineering" was peer-reviewed at PLOS Biology as a Methods and Resorces paper. Please also accept my apologies for the time it has taken us to provide you with a decision. The manuscript has now been evaluated by the PLOS Biology editors, an Academic Editor with relevant expertise, and by two independent reviewers. 

The reviews are attached below. As you will see, the reviewers find the tool you developed novel and important for the field, but they also raise several issues that would need to be address before we consider the manuscript for publication. Both reviewers raise concerns regarding the use of the term ‘translocation efficiency’ as it is not measured directly in the study, and Reviewer 2 thinks you should perform additional experiments showing the actual translocation efficiency. This reviewer also thinks you should acknowledge some of the limitations in the text, such as the use of a transient heterologous system in tobacco.

In light of the reviews, we would like to invite you to revise the work to thoroughly address the reviewers' reports. Given the extent of revision needed, we cannot make a decision about publication until we have seen the revised manuscript and your response to the reviewers' comments. Your revised manuscript is likely to be sent for further evaluation by all or a subset of the reviewers.

**IMPORTANT - SUBMITTING YOUR REVISION**

3. Resubmission Checklist

a) *PLOS Data Policy*

b) *Published Peer Review*

d) *Blurb*

Please also provide a blurb which (if accepted) will be included in our weekly and monthly Electronic Table of Contents, sent out to readers of PLOS Biology, and may be used to promote your article in social media. The blurb should be about 30-40 words long and is subject to editorial changes. It should, without exaggeration, entice people to read your manuscript. It should not be redundant with the title and should not contain acronyms or abbreviations. For examples, view our author guidelines: https://journals.plos.org/plosbiology/s/revising-your-manuscript#loc-blurb

Sincerely,

Ines

--

Ines Alvarez-Garcia, PhD

Senior Editor

PLOS Biology

Reviewers' comments

Rev. 1:

This manuscript describes the screening of a large number of chloroplast transit peptides (cTPs) for their effectiveness at driving the accumulation of GFP in chloroplasts. One particularly effective cTP (from the chloroplast ribosomal protein L35) was discovered and then further analysed via deletion analysis (unsurprisingly, the full-length cTP was the most effective). Importantly, the authors demonstrated that the effectiveness of the L35 cTP was not limited to GFP fusions by showing that other cargo proteins also accumulated to higher levels within chloroplasts. The findings in this manuscript will be of great practical interest to many plant biologists and biotechnologists, as targeting proteins to chloroplasts is a common requirement of many projects.

Major point

Much of the text, as written, appears to assume that the higher level of GFP fluorescence in chloroplasts observed when the L35 cTP is used is due to more efficient translocation of the fusion protein into chloroplasts. However, the fluorescence in this end-point assay depends quantitatively on many other factors that may be affected by the sequence of the cTP. These factors include, for example, mRNA stability, translational efficiency, interaction with cytosolic chaperones, protein folding (or unfolding), protein stability, and the rate of GFP fluorophore formation. The only direct measure of the rate of translocation into chloroplasts (Fig 3) failed to provide any evidence for improved import mediated by the L35 cTP. So I think the authors should be much more careful about the interpretation of their results, and rephrase their text to remove any implicit or explicit assumptions about the mechanism by which the L35 cTP acts to boost the accumulation of cargo proteins in chloroplasts. It could well be that more GFP is found in chloroplasts when expressed as an L35 cTP fusion because more preprotein is synthesized in the first place, and not because it is more efficiently translocated. I emphasize that this is a pervasive issue in the text — there are dozens of places where it is written that the L35 cTP improves translocation, despite the lack of any evidence that it does so.

Minor points

p3, lines 60-61: 'N-terminal cTPs are usually located 20 to >100 amino acids upstream of preproteins'. The cTP is part of the pre-protein, and thus is not 'upstream' of it. This should be re-phrased, e.g. 'N-terminal cTPs generally encompass the first 20 to 100 amino acids of the preproteins'.

p4, lines 86-87: 'However, to date, a comparison of the translocation efficiencies of cTPs has not been performed'. This statement is simply not true. Many studies comparing different cTPs have been performed by different groups. They may not have included as many CTPs as were tested here, or the focus of the study may have been different, but it would be unfair not to mention some here. As a couple of examples:

Chiung-Chih Chu, Krishna Swamy, Hsou-min Li, Tissue-Specific Regulation of Plastid Protein Import via Transit-Peptide Motifs, The Plant Cell, Volume 32, Issue 4, April 2020, Pages 1204-1217, https://doi.org/10.1105/tpc.19.00702

Eseverri Álvaro, Baysal Can, Medina Vicente, Capell Teresa, Christou Paul, Rubio Luis M., Caro Elena. Transit Peptides From Photosynthesis-Related Proteins Mediate Import of a Marker Protein Into Different Plastid Types and Within Different Species. Frontiers in Plant Science 11 (2020) DOI=10.3389/fpls.2020.560701

p6, lines 131-132: 'we did not observe sequence similarity in cleavage sites among the predicted cTPs and tTPs'. As Fig S1 shows, there is clear conservation of amino acids around cleavage sites, as one would expect. So I do not understand why the authors say the contrary in the main text. Incidentally, in Fig S1, the authors should indicate exactly where the cleavage sites are, in panels B and C.

p7, lines 148-150: 'Despite their alternative localizations in mitochondria and cytosol, cTP-GFPs targeted to chloroplasts exhibited higher GFP fluorescence than their cytosolic

counterparts'. The GFP signal has been normalised with respect to the chlorophyll autofluorescence signal. This seems reasonable for GFP signals within chloroplasts, as the two signals should be largely co-located, but not for GFP signals in other cellular compartments, where there is no reason to expect a strong correlation with chlorophyll autofluorescence. Indeed they may even be anti-correlated — a focal plane through a part of the cell cytoplasm with many chloroplasts may have relatively little cytosol as the two compartments are mutually exclusive. I don't think the authors can meaningfully compare GFP fluorescence signals between different compartments using this method. Comparing signals on western blots would probably be better.

p10, lines 223-224: 'Chloroplast-targeting peptides (cTPs) are evolutionarily designed to interact with the chloroplast import machinery'. As I'm sure the authors are aware, evolution does not design anything, so this is inappropriate phrasing. I don't think the sentence is necessary, I recommend deleting it.

Figure 5: In these experiments, there is the theoretical possibility of inducing silencing of the endogenous L35 gene, leading to potential effects on chloroplast biogenesis and metabolism. Can the authors rule out this possibility? I would like to see some discussion of this, and ideally some evidence showing that this is not an issue.

Rev. 2:

The paper titled "Identification of a Highly Efficient Chloroplast-Targeting Peptide for Plastid Engineering" authored by Thagun et al. aims to identify an optimal chloroplast targeting peptide (CTP) for biotechnological applications. The study involves the evaluation of 89 CTP candidates selected from Arabidopsis chloroplast proteins, wherein these candidates were fused with GFP in a transient expression system using tobacco plants. Among these candidates, the CTP from At2g24090 exhibited the best performance. Subsequently, this top-performing CTP was further assessed by fusion with enzymes to demonstrate its enhanced accumulation and efficient chloroplast localization for the selected enzymes.

While the proposed CTPs hold promise for advancing chloroplast biotechnology, there are several major concerns that require attention for a more thorough and clear explanation. Please consider the following suggestions for revision.

Major comments

1) The term "translocation efficiency" and its associated explanations should be carefully reviewed and revised throughout the manuscript. The authors primarily assessed the accumulation of fused proteins in chloroplasts but did not directly measure "translocation efficiency." Translocation efficiency typically involves determining the ratio of protein content within chloroplasts to the total protein content or examining the speed of protein translocation from the cytosol to the chloroplast. It is suggested that the observed effective accumulation of proteins using the CTP from At2g20490 may be attributed to the efficiency of transcription and translation processes rather than the effects of "translocation efficiency." Therefore, the manuscript should provide a more accurate and clear representation of the experimental results and their implications.

2) The selection of Arabidopsis CTPs for promoting protein accumulation in chloroplasts was conducted using a transient heterologous system in tobacco. While this approach may be suitable for initial screening, it may not necessarily reflect the effectiveness of these CTPs when applied within the native plant context. The manuscript should acknowledge the limitations of this method and clearly state that further validation or testing within the intended plant species is necessary.

In practical bioengineering applications, the key objective often revolves around achieving higher protein accumulation within chloroplasts, and this should be emphasized and explained more comprehensively in the manuscript. Revisions according to these two major points would enhance the clarity and suitability of the paper for publication.

Minor

Line 149, "higher GFP fluorescence than their cytosol counterparts" would be

"Higher GFP fluorescence than their other counterparts in hte cytoplasm"

---

## [Decision Letter · Decision Letter 2]

18 Jul 2024

Dear Dr Numata,

Thank you for your patience while we considered your revised manuscript entitled "Identification of a highly efficient chloroplast-targeting peptide for plastid engineering" for publication as a Methods and Resources at PLOS Biology. This revised version of your manuscript has been evaluated by the PLOS Biology editors, the Academic Editor and the two original reviewers.

Based on the reviews (attached below), we are likely to accept this manuscript for publication, provided you satisfactorily address the data and other policy-related requests stated below.

We expect to receive your revised manuscript within two weeks. 

*Published Peer Review History*

*Press*

Sincerely,

Ines

--

Ines Alvarez-Garcia, PhD

Senior Editor

PLOS Biology

Fig. 1B-D, G-I; Fig. 2B, C, G; Fig. 3F-H; Fig. 4A, B, D, F-I; Fig. 6D-G, I-N; Fig. 7E, G, I-K; Fig. S1A; Fig. S9A, B; Fig. S11A, B; Fig. S14; Fig. S18C; Fig. S19C; Fig. S20B, C; Fig. S21A-D and Fig. S22C

CODE POLICY

We require the original, uncropped and minimally adjusted images supporting all blot and gel results reported in an article's figures or Supporting Information files. We will require these files before a manuscript can be accepted so please prepare and upload them now. Please carefully read our guidelines for how to prepare and upload this data: https://journals.plos.org/plosbiology/s/figures#loc-blot-and-gel-reporting-requirements

Reviewers' comments

Rev. 1:

Thank you for making such a strong effort to respond the reviewers' comments. I am convinced by the new data that the primary impact of switching the cTP sequence is on import efficiency and not, for example, on expression level. All the criticisms I had of the first version of the manuscript have been more than adequately addressed.

Rev. 2:

I am satisfied with the author's response and revision.

---

## [Editor Report · Decision Letter 3]

3 Aug 2024

Dear Dr Numata,

Thank you for the submission of your revised Methods and Resources entitled "Identification of a highly efficient chloroplast-targeting peptide for plastid engineering" for publication in PLOS Biology. On behalf of my colleagues and the Academic Editor, Mark Estelle, I am delighted to let you know that we can in principle accept your manuscript for publication, provided you address any remaining formatting and reporting issues. These will be detailed in an email you should receive within 2-3 business days from our colleagues in the journal operations team; no action is required from you until then. Please note that we will not be able to formally accept your manuscript and schedule it for publication until you have completed any requested changes.

PRESS

Sincerely, 

Ines

--

Ines Alvarez-Garcia, PhD

Senior Editor

PLOS Biology
